# Bandit Quickest Changepoint Detection

**Aditya Gopalan**
Electrical Communication Engineering
Indian Institute of Science, Bengaluru 560012
`aditya@iisc.ac.in`

**Braghadeesh Lakshminarayanan**
Electrical Communication Engineering
Indian Institute of Science, Bengaluru 560012
`braghadeesh94@gmail.com`

**Venkatesh Saligrama**
Electrical and Computer Engineering
Boston University, Boston, MA 02215
`srv@bu.edu`

## Abstract

Many industrial and security applications employ a suite of sensors for detecting abrupt changes in temporal behavior patterns. These abrupt changes typically manifest locally, rendering only a small subset of sensors informative. Continuous monitoring of every sensor can be expensive due to resource constraints, and serves as a motivation for the bandit quickest changepoint detection problem, where sensing actions (or sensors) are sequentially chosen, and only measurements corresponding to chosen actions are observed. We derive an information-theoretic lower bound on the detection delay for a general class of finitely parameterized probability distributions. We then propose a computationally efficient online sensing scheme, which seamlessly balances the need for exploration of different sensing options with exploitation of querying informative actions. We derive expected delay bounds for the proposed scheme and show that these bounds match our information-theoretic lower bounds at low false alarm rates, establishing optimality of the proposed method. We then perform a number of experiments on synthetic and real datasets demonstrating the effectiveness of our proposed method.

## 1 Introduction

We propose a framework for bandit[1] quickest change detection (BQCD). Specifically, we are given a multi-dimensional online data stream, which can be sequentially probed by means of actions belonging to an action set (for instance, only a few among all of the data stream components can be observed at any time, or a linear combination can be acquired). The online data stream can exhibit abrupt *statistical* changes, at any time, and only among a few arbitrary components. Our task is to sequentially probe the data stream by adaptively choosing valid actions based on past bandit[2] observations associated with past actions. Our objective is to detect a change, if it has happened, with minimum detection delay.

*Example Scenarios.* Surveillance systems [HC11] are equipped with a suite of sensors that can be switched and steered to focus attention on any target or location over a physical landscape (see Fig. 1) to detect abrupt changes at any location. On the other hand, sensor suites are resource limited, and only a limited subset, among all the locations, can be probed at any time. As such,

---

[1] By 'bandit' we generally mean adaptive sampling with partial information.

[2] This is different from classical quickest change detection [TNB14], where all of the multiple streams are observed at each time, and the only adaptive decision is to declare whether or not a change has happened.

35th Conference on Neural Information Processing Systems (NeurIPS 2021).

we face a fundamental *dilemma*: Focusing attention at any one location can compromise change detection at other locations. Although we describe a scenario in surveillance, the problem of BQCD is general and arises in intrusion detection [Bas99], social networks [Vis+14], disease outbreaks and epidemics [QSC14; YSK15], manufacturing processes [Pur+19a; Din+06], energy limited sensor networks[OGR10; ES10], and vital health monitoring [Vil+17].

In this paper, we derive a fundamental characterization for the delay performance of BQCD, and make explicit, the fundamental tradeoff between delay and false-alarm in the low-false alarm regime. We specifically make the following contributions.

**Information-theoretic Lower Bound.** We prove a lower bound on the expected detection delay that any BQCD algorithm must suffer at a fixed false alarm rate. The lower bound exhibits a fundamental tradeoff between early stopping (false-alarm) and detection delay (time to detect abrupt change). It offers the key insight that the quickest way to detect a change, at any false-alarm rate, is by playing the 'most informative action', of an 'oracle' who a priori knows the post-change distribution. This suggests that to quickly identify changepoints, we must direct our effort towards rapidly identifying informative actions. On a technical level, we develop a change-of-measure argument for nonstationary, adaptive change detection, that allows for relating the divergences between random trajectories until stopping to the divergence of probability laws under each action for any two problem instances.

**ε-Greedy Change Detector (ε-GCD).** We propose ε-GCD, which, at a high level, uses a small amount of forced exploration to identify informative actions. The forced exploration allows for rapid convergence towards informative actions, and playing these actions minimizes detection delay. Our ε-GCD is based on the generalized max-likelihood/likelihood ratio principle which is utilized to estimate parametric changes. To prove detection delay bounds we draw upon key insights of ε-GCD. We first interpret the scheme in terms of competing parallel 'queues', where each queue corresponds to a candidate post-change parameter, collects 'arrivals' which are log-likelihood ratios of observations, and cannot go negative. The true parameter is the queue which enjoys the highest growth rate after a change, and the detection delay is the time required for it to dominate and become the 'longest queue'. The dynamics of the queues can be related to nonstationary random walks with drifts. Using these insights, we prove that the expected detection delay of ε-GCD at low false alarm rates mirrors our information theoretic lower bounds thus establishing optimality of our method.

**Experiments.** We perform numerical experiments of ε-GCD on synthetic and real datasets and show that under variations of changepoints, anomalies, and action sets, we realize gains due to adaptive sensing.

## 2 Related Work

Classical quickest change detection, dating back to the pioneering works of Page [Pag54] and Lorden [Lor71], studies the problem of deciding when to stop (akin to declaring a change) while sequentially observing a data stream. In this context, the CUSUM algorithm [Pag54], in

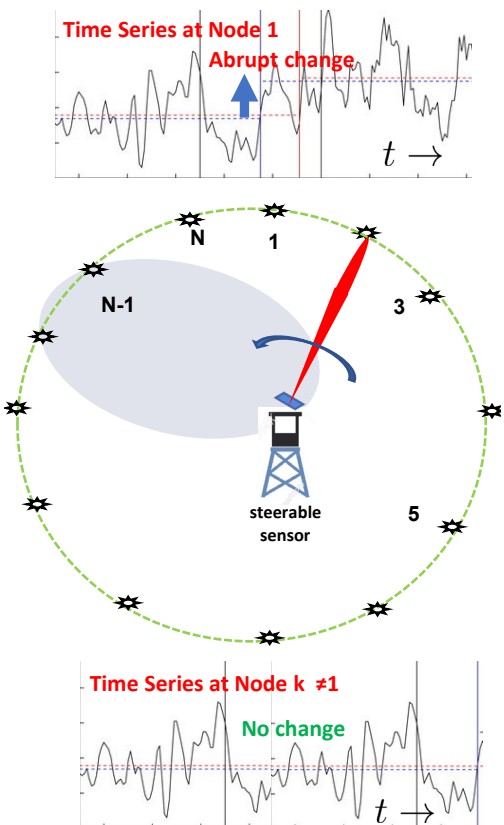

Figure 1: Example BQCD Scenario. There are $N$ spatial-locations that are monitored. The steerable sensor is capable of rapidly switching modes–grey and red beams as shown–and collect aggregated signal from multiple locations at a low signal-to-noise ratio (SNR) or focus attention narrowly on a single location at high SNR. Locations can exhibit abrupt change, which manifests as the depicted time-series (see top). Due to resource constraints, at any time, only a limited set of locations can be observed with high SNR. BQCD aims to learn a policy to adaptively steer and switch modes to different locations and detect abrupt changes if any, quickly.

which a change is announced as soon as a likelihood ratio statistic exceeds a threshold, has been shown to enjoy optimal performance.

Our work is motivated by constraints on data collection in multi-stream time-series data, necessitating the design of an adaptive sampling rule in addition to the stopping rule. Similar to our work, Zhang and Mei [ZM20] also propose a sequential sampling method for quickest change detection based on the well-known Shiryaev-Roberts-Pollak scheme (see, e.g., [BN93]). However, their theoretical analysis is rather limited in its scope. In particular, they argue "in principle" that their method will control false alarms for sufficiently large thresholds, and will *eventually* (i.e., after infinite time) choose sensors where changes manifest, without any explicit characterization of detection delays. In contrast, we derive fundamental detection delay under a false alarm rate constraint, thus fully characterizing the BQCD problem. Additionally, their framework is not general enough to handle correlated or structured anomalies, which is a significant aspect of both our theoretical and experimental investigations.

In other related work, Gundersen et al [Gun+21] propose to extend the framework of Bayesian Online Change Detection (BOCD) [AM07] to incorporate costs in making decisions for real-time data acquisition in multi-fidelity sensing scenarios. Different from this perspective, our approach is frequentist, and does not impose action-specific costs and distributional constraints on the underlying latent parameters or on the changepoints. Furthermore, we derive finite-time performance bounds, which is not a focus of this work.

## 3 Problem Setup

**General Setup.** We consider the following discrete-time model for the bandit quickest changepoint detection problem. At each decision round $t \in \mathbb{N} := \{1, 2, \ldots\}$, the parameter $\theta_t$ governs the distribution of the observation taken in that round. Let $\nu \in \mathbb{N}$ denote the round at which the change in behavior occurs, so that $\theta_t = \theta_0$ for $t < \nu$ and $\theta_t = \theta^* \in \Theta^{(1)}$ for $t \geq \nu$. At each round $t$, a learner decides to either (1) stop and output that a change has taken place, denoted by the random variable $U_t = 1$, or (2) continue (denoted by $U_t = 0$) and play an action or sensing decision $A_t$ from an action set $\mathcal{A}$. Upon playing this action the learner obtains an observation $X_t$, sampled independent of the past, from a probability distribution $\mathbb{P}_{\theta_t} [\cdot | A_t]$, which depends on both the current system parameter $\theta_t$ and the current action $A_t$. The stopping decision and sensing action ($U_t$ and $A_t$) are assumed to be chosen in a causal manner, i.e., depending on all past information $U_0, A_1, X_1, U_1, \ldots, A_{t-1}, X_{t-1}$, along with potentially independent internal randomness. The learner is aware, a priori, of the pre-change parameter $\theta_0$, the post-change parameter set $\Theta^{(1)}$ and the observation distribution structure, i.e., $\{\mathbb{P}_\theta [\cdot | a]\}_{\theta, a}$ with $a \in \mathcal{A}$ denotes fixed actions. Crucially, the change time $\nu$ and the specific post-change parameter $\theta^*$ are not known in advance and must be 'learnt' in order to perform well.

The stopping time of the learner is defined to be $\tau = \min\{t \geq 0 : U_t = 1\}$, and the main performance metric that we are interested in is the detection delay, which is the random variable $(\tau - \nu)^+ = \max\{\tau - \nu, 0\}$. Specifically, we are interested in designing sampling and stopping rules so that (a) when no change occurs ($\nu = \infty$), $\tau$ should be at least a specified time delay with high probability, which is a measure of false alarm rate, and (b) when a change occurs ($\nu < \infty$), the expected detection delay should be small implying quick detection of change.

**Illustrative Example.** For illustration, we elaborate on the scenario depicted in Fig. 1. We consider $N$ 'locations' as shown, and the observation at location $j$ at time $t$ follows:

$$Y_j(t) = \theta \mathbb{1}_{[t \geq \nu] \wedge [j \in S]} + W_j(t), \ t \in \mathbb{Z}^+, \ j \in [N],$$

where $\{W_j(t)\}_{j,t}$ is a collection of standard normal random variables and $\mathbb{1}_{[\cdot]}$ denotes the indicator function. This is a temporal extension of the model in [QSC14]. The model with the parameter $\theta \neq 0$ signifies an elevated mean after time $t \geq \nu \in \mathbb{Z}^+$ at locations $S \subseteq [N]$. The learner is agnostic to $\theta$, the time $\nu$ as well as which set $S$ among a family of known subsets $\mathcal{S} \subseteq \{0, 1\}^N$ exhibits elevated activity. The learner at time $t$ can probe locations by means of sensing actions $A_t \in \mathcal{A} \subset \{0, 1\}^N$, and observe a scalar measurement, $X_t \sim P_{\theta_t}(\cdot \mid A_t) \sim \mathcal{N}\left(\theta_t \frac{|A_t \cap S|}{\sqrt{|A_t|}}, 1\right)$, with $\theta_t = 0$ for $t < \nu$ and for $t \geq \nu$, $\theta_t = \theta \in \Theta^{(1)} = \mathbb{R} \setminus \{0\}$. The normalization factor $\sqrt{|A_t|}$ expresses the fact that the same SNR is maintained across different probes $A_t$ (see [QSC14]).

*General (Structured) Scenarios.* Depending on the application, locations maybe endowed with a neighborhood network structure, and change maybe observable in a $K$-neighborhood of the target location, and in such cases, $\mathcal{S}$ are constrained to be $K$-neighborly sets. Similarly, to allow for diverse probing modes, we could allow the action set to include actions that aggregate signal across multiple locations.

**Notation.** For a post-change parameter $\theta \in \Theta^{(1)}$, we use $a^*(\theta)$ to denote the most *informative* action for $\theta$ w.r.t. $\theta^{(0)}$, i.e., $a^*(\theta) \in \operatorname{argmax}_{a \in \mathcal{A}} D\left(\mathbb{P}_\theta\left[\cdot|a\right] || \mathbb{P}_{\theta_0}\left[\cdot|a\right]\right)$, where $D\left(\cdot||\cdot\right)$ stands for the Kullback-Leibler divergence. To lighten notation, in the sequel we often abbreviate the distribution $\mathbb{P}_\theta\left[\cdot|a\right]$ to $\theta(a)$, and thus $D\left(\mathbb{P}_\theta\left[\cdot|a\right] || \mathbb{P}_{\theta_0}\left[\cdot|a\right]\right)$ to $D\left(\theta(a)||\theta_0(a)\right)$. The simultaneous subscript-superscript notation $z_i^j$ is often used to represent the sequence $(z_i, z_{i+1}, \dots, z_j)$. For any $\theta \in \Theta^{(1)}$ and $n \in \mathbb{N}$, we use $\mathbb{P}^{(n,\theta)}$ to denote the probability measure on the process $U_0, A_1, X_1, U_1, A_2, X_2, \dots$ induced by the learner's decisions, when the pre-change parameter $\theta_0$ changes to $\theta$ at time $n$. We also denote by $\mathbb{P}^{(\infty)}$ the probability measure as above under the no-change situation ($\nu = \infty$).

We define the filtration $(\mathcal{F}_t)_{t \geq 0}$ to be the $\sigma$-algebra generated by all the past random variables up to time $t$, i.e., $\mathcal{F}_t = \sigma(\{(A_s, X_s), 0 \leq s \leq t)\})$. An (admissible) change detector is an algorithm for which the stopping decision $U_t \in \{0, 1\}$ and action $A_t \in \mathcal{A}$ are $\mathcal{F}_{t-1}$-measurable at all rounds $t$.

**Objective Specification.** Our BQCD framework involves a false alarm rate constraint specified by a tuple $(m, \alpha)$, where $m \in \mathbb{Z}^+$ denotes an a priori fixed time such that the probability of an admissible change detector stopping before time $m$ under the null hypothesis (no change) is at most $\alpha \in (0, 1)$, namely, $\mathbb{P}^{(\infty)}\left[\tau < m\right] \leq \alpha$. This kind of false alarm duration specification for the stopping time ($\geq m$ with probability $\geq 1 - \alpha$) has been commonly employed in the literature on classical (non-adaptive) change detection (see e.g., [Lai98]), to ensure that the comparative performance rules out early triggering algorithms. Let $\mathcal{D}_{m,\alpha}$ be the class of admissible change detectors with the $(m, \alpha)$ property stated above. As such, our goal is to find detectors in $\mathcal{D}_{m,\alpha}$ that minimize the detection delay when a change indeed occurs.

**Informal Statement of Our Results.** Our main result is a sharp "optimal characterization" for expected detection delay in the limit of vanishing false alarms ($\alpha \to 0, m \to \infty$). The result, stated below, involves the new $\epsilon$-Greedy-change-detector ($\epsilon$-GCD) algorithm in which, in each round $t$, we choose an action uniformly at random from the set $\mathcal{A}$ with probability $\epsilon$, or choose an action that is 'most informative' for an estimated post-change parameter with probability $1 - \epsilon$. **Objective Specification.** Our BQCD framework involves a false alarm rate constraint specified by a tuple $(m, \alpha)$, where $m \in \mathbb{Z}^+$ denotes an a priori fixed time such that the probability of an admissible change detector stopping before time $m$ under the null hypothesis (no change) is at most $\alpha \in (0, 1)$, namely, $\mathbb{P}^{(\infty)}\left[\tau < m\right] \leq \alpha$. This kind of

**Theorem 1** (informal)**.** *Consider the limiting situation as $\alpha \to 0$ and $m = \omega\left(\log \frac{1}{\alpha}\right) \to \infty$. Then, any admissible change detector in $\mathcal{D}_{m,\alpha}$ must suffer*

$$\liminf_{\alpha \to 0} \frac{\mathbb{E}^{(\nu,\theta^*)}[(\tau - \nu)^+]}{\log \frac{1}{\alpha}} \geq \frac{1}{\max_a D(\theta^*(a)||\theta^{(0)}(a))}.$$

*On the other hand, the $\epsilon$-GCD algorithm, suitably tuned to be in $\mathcal{D}_{m,\alpha}$, achieves*

$$\limsup_{\alpha \to 0} \frac{\mathbb{E}^{(\nu,\theta^*)}[(\tau - \nu)^+]}{\log \frac{1}{\alpha}} \leq \frac{8}{(1 - \epsilon)} \frac{1}{\max_a D(\theta^*(a)||\theta^{(0)}(a))}.$$

## 4 Fundamental lower bounds on bandit change detection performance

Before embarking upon the design of bandit changepoint algorithms, it is worth understanding what the limits of bandit change detection performance are, due to the stochastic and partial nature of the problem's information structure. We expose in this section a universal lower bound on the detection delay of *any* bandit changepoint algorithm with a false alarm rate above a given level.

**Theorem 2.** *Let $0 < \alpha \leq \frac{1}{10}$ and $m \geq 1$. For any bandit changepoint algorithm satisfying*
$$\mathbb{P}^{(\infty)}\left[\tau < m\right] \leq \alpha, \text{ we have } \mathbb{E}^{(1,\theta^*)}\left[\tau\right] \geq \min\left\{\frac{\frac{1}{20}\log \frac{1}{\alpha}}{\max_{a \in \mathcal{A}} D\left(\theta^*(a)||\theta^{(0)}(a)\right)}, \frac{m}{2}\right\} \quad \forall \theta^* \in \Theta^{(1)}.$$

The result implies that in the 'small' false alarm rate ($\alpha$) regime with 'large' $m = \omega(\log(1/\alpha))$, e.g., $m = \log^2(1/\alpha)$, we have that any algorithm meeting the false alarm rate property $\mathbb{P}^{(\infty)}\left[\tau \geq m\right] \geq$

$1 - \alpha$ must suffer a detection delay at least $\Omega\left(\frac{\log \frac{1}{\alpha}}{\max_{a \in \mathcal{A}} D\left(\theta^*(a) \| \theta^{(0)}(a)\right)}\right)$ when a change occurs at time 1 (a similar, 'anytime' lower bound holds for detection delay at $\nu \in \mathbb{N}$, see appendix).

**False alarm-detection Delay Tradeoff.** Theorem 2 shows that a basic tradeoff exists between the false alarm rate or early stopping rate in case of no change, on one hand, and the detection delay after a true change occurs, on the other. Specifically, it is impossible to stop 'too early', i.e., before time $\Omega\left(\frac{\log \frac{1}{\alpha}}{\max_{a \in \mathcal{A}} D\left(\theta^*(a) \| \theta^{(0)}(a)\right)}\right)$, after a true change if one wishes to stop 'too late' under no change, i.e., $\mathbb{P}^{(\infty)}[\tau \geq m] \geq 1 - \alpha$. We also note that when there is no adaptive sampling of actions (i.e., $|\mathcal{A}| = 1$), then the lower bound reduces to the form of a a standard lower bound for the classical changepoint detection problem [Lai98].

**Information Structure.** This is perhaps the most valuable implication of the lower bound. The quantity $D\left(\theta^*(a) \| \theta^{(0)}(a)\right)$, in the denominator, can be interpreted as the amount of information that playing an action $a$ provides (on average) in order to detect a change from $\theta_0$ to $\theta^*$. Consequently, the quickest way to detect a change is by playing the 'most informative action', $\operatorname{argmax}_{a \in \mathcal{A}} D\left(\theta^*(a) \| \theta^{(0)}(a)\right)$. This can be viewed as the sampling strategy of an 'oracle' who knows the post-change parameter $\theta^*$ in advance. Unfortunately, this information is not known a priori for a causal algorithm. However, we will see that this can in fact be learnt along the trajectory and the lower bound can be attained order-wise by a suitable learning algorithm that we propose.

**Proof Sketch for Theorem 2.** The main idea is a measure change argument, adapted to our non-stationary change point setting, from literature on sample complexity bounds for bandit best arm identification [GK16]. On one hand, if the algorithm in consideration is very 'lazy' to begin with, i.e., $\mathbb{P}^{(1, \theta^*)}[\tau \geq m]$ is at least a constant, say $1/2$, then we immediately get $\mathbb{E}^{(1, \theta^*)}[\tau] \geq \frac{1}{2} \cdot m$. On the other hand, if the algorithm is 'active', i.e., $\mathbb{P}^{(1, \theta^*)}[\tau < m] \geq 1/2$, then the KL divergence between the laws of the indicator random variable $\mathbf{1}\{\tau < m\}$ in the instances $(1, \theta^*)$ and $(\infty, \cdot)$ is 'large' (at least a constant times $\log(1/\alpha)$) owing to the hypothesis that $\mathbb{P}^{(\infty)}[\tau < m]$ is 'small' (at most $\alpha$). But by the data processing inequality, this divergence is bounded above by the divergence between the distributions of the entire trajectory $U_0, A_1, X_1, U_1, A_2, X_2, \ldots$ under the two instances, which can be seen to be equivalent to the quantity $\sum_{a \in \mathcal{A}} D\left(\theta^*(a) \| \theta(a)\right) \mathbb{E}^{(1, \theta^*)}[N_\tau(a)]$, and which is further bounded above by $\mathbb{E}^{(1, \theta^*)}[\tau] \max_{a \in \mathcal{A}} D\left(\theta^*(a) \| \theta(a)\right)$. See appendix for a complete proof.

## 5  $\epsilon$-GCD Algorithm

We describe the $\epsilon$-GCD adaptive sensing algorithm for the bandit quickest change detection problem. The lower bound in Section 4 suggests that it is beneficial to infer the target post-change parameter, so that playing the most informative action for it can yield the best possible detection delay performance. This is the key principle underlying the design of the $\epsilon$-GCD algorithm (Algorithm 1).

At a high level, $\epsilon$-GCD uses a small amount of forced exploration along with 'greedy' exploitation to play sensing actions. Specifically, it computes, at each round $t$, a maximum likelihood estimate (MLE) of the post-change parameter based on the generalized likelihood ratio test (GLRT) principle. It then plays either a randomly chosen action, if the current slot is an exploration slot, or a 'greedy', i.e., most informative, action for the estimated post-change parameter, if it is an exploitation slot.

The MLE of the post-change parameter, in round $t$, admits an interpretation as the longest 'queue' corresponding to some parameter in $\Theta^{(1)}$. To see this, notice that the MLE for the pair $(\nu, \theta^*)$, given all previous data in exploration rounds can be written as $\operatorname{argmax}_{\theta \in \Theta^{(1)}, 1 \leq v \leq t} \prod_{\ell=1}^{v-1} f_{\theta_0}(X_\ell | A_\ell)^{E_\ell} \prod_{\ell=v}^{t-1} f_\theta(X_\ell | A_\ell)^{E_\ell} = \operatorname{argmax}_{\theta \in \Theta^{(1)}, 1 \leq v \leq t} \sum_{\ell=v}^{t-1} E_\ell \log \frac{f_\theta(X_\ell | A_\ell)}{f_{\theta_0}(X_\ell | A_\ell)}$, with $f_\theta(\cdot | a)$ taken to be the probability density or mass function of the distribution $\mathbb{P}_\theta(\cdot | A_\ell)$ (assuming one exists). Observe now that for each candidate post-change parameter $\theta$, the inner maximum over $v$, $Q_t^{(1)}(\theta) := \operatorname{argmax}_{1 \leq v \leq t} \sum_{\ell=v}^{t-1} E_\ell \log \frac{f_\theta(X_\ell | A_\ell)}{f_{\theta_0}(X_\ell | A_\ell)}$, evolves as a 'queue'[3] with arriving work $E_\ell \log \frac{f_\theta(X_\ell | A_\ell)}{f_{\theta_0}(X_\ell | A_\ell)}$ at time slot $\ell$; in other words, $Q_{t+1}^{(1)}(\theta) = \left(Q_t^{(1)}(\theta) + E_t \log \frac{f_\theta(X_t | A_t)}{f_{\theta_0}(X_t | A_t)}\right)^+$.

---

[3]This is also known as the Lindley recursion equation in queueing theory.

We define the algorithm using general 'processing' functions $g_\theta$ in place of the log-likelihood ratios $\log \frac{f_\theta}{f_{\theta_0}}$ above. Broadly speaking, the functions $g$ should ideally be chosen with the hope that (a) $g_\theta(X_\ell, A_\ell)$ is negative in expectation before the change time ($\ell < \nu$), and (b) $g_\theta(X_\ell, A_\ell)$ is large and positive in expectation for $\theta = \theta^*$, the true change parameter, after the change ($\ell \geq \nu$).

The stopping rule that $\epsilon$-GCD uses is based on the generalized likelihood ratio (GLR)-type statistic. It is the largest of an ensemble of evolving exploitation-data queues $Q_t^{(2)}(\theta)$, which mirrors the definition of $Q_t^{(1)}(\theta)$ but with $1 - E_t$ instead of $E_t$.

## 5.1 Theoretical guarantees for the $\epsilon$-GCD algorithm

In this section, we present and discuss theoretical guarantees on the false alarm rate and detection delay performance of the $\epsilon$-GCD algorithm of Section 5.

We make the following assumptions on the parameter space and observation distributions in order to derive performance bounds for the algorithm.

**Assumption 1.** *The post-change parameter set $\Theta^{(1)}$ is finite, i.e., $|\Theta^{(1)}| < \infty$.*

This assumption is made primarily for ease of analysis, whereas the algorithm is defined even for arbitrary parameter spaces. Specifically, it allows for easy control of the fluctuations of an ensemble of (drifting) random walks via a union bound over the parameter set. While we believe that this can be relaxed to handle general parameter spaces via appropriate netting or chaining arguments, this chiefly technical task is left to future investigation.

**Assumption 2.** *Every post-change parameter is detectable by some action, i.e., $\forall \theta \in \Theta^{(1)} \, \exists a \in \mathcal{A} : D\left(\theta(a)||\theta^{(0)}(a)\right) > 0$. Moreover, any two post-change parameters are separable by some action, i.e., $\forall \theta, \theta' \in \Theta^{(1)} \, \exists a \in \mathcal{A} : D\left(\theta(a)||\theta'(a)\right) > 0$.*

This is a basic identifiability requirement of the setting, without which some parameter changes could be completely undetectable. Put differently, one can only hope to detect changes that the sensing set can tease apart.

**Assumption 3** (Bounded marginal KL divergences)**.** *There is a constant $D_{\max}$ satisfying $D\left(\theta_1(a)||\theta_2(a)\right) \leq D_{\max}$ for each $a \in \mathcal{A}$, $\theta_1, \theta_2, \in \Theta^{(1)}$.*

---

**Algorithm 1** $\epsilon$-GCD

1: **Input:** $\epsilon \in [0, 1]$, $\theta_0$, $\Theta^{(1)}$, $\beta \geq 1$, $\pi$, Observation function $g_\theta(x, a) \, \forall (\theta, x, a) \in \Theta^{(1)} \times \mathcal{X} \times \mathcal{A}$.

2: **Init:** $Q_1^{(1)}(\theta) \leftarrow 0$, $Q_1^{(2)}(\theta) \leftarrow 0 \, \forall \theta \in \Theta^{(1)}$ {CUSUM statistics for exploration and exploitation per $\theta$}

3: **for** round $t = 1, 2, 3, \ldots$ **do**

4:     **if** $\max_{\theta \in \Theta^{(1)}} Q_t^{(2)}(\theta) \geq \log \beta$ **then**

5:         **break**       {Stop sampling and exit}

6:     **end if**

7:     Sample $E_t \sim \text{Ber}(\epsilon)$ independently

8:     **if** $E_t == 1$ **then**

9:         Play action $A_t \sim \pi$ independently   {Explore}

10:         Get observation $X_t$

11:         Set $\forall \theta \in \Theta^{(1)} : Q_{t+1}^{(1)}(\theta) \leftarrow \left(Q_t^{(1)}(\theta) + g_\theta(X_t, A_t)\right)^+$   {Update exploration CUSUM statistics}

12:     **else**

13:         Compute $\hat{\theta}_t = \text{argmax}_{\theta \in \Theta^{(1)}} Q_t^{(1)}(\theta)$ {Most likely post-change distribution based on exploration data}

14:         Play action $A_t = \text{argmax}_{a \in \mathcal{A}} D\left(\theta^{(0)}(a)||\hat{\theta}_t(a)\right)$

15:         Get observation $X_t$

16:         Set $\forall \theta \in \Theta^{(1)} : Q_{t+1}^{(2)}(\theta) \leftarrow \left(Q_t^{(2)}(\theta) + g_\theta(X_t, A_t)\right)^+$   {Update exploitation CUSUM statistics}

17:     **end if**

18: **end for**

---

This assumption is easily met, for instance, if all log likelihood ratios are bounded, or if the observations are modelled as Gaussians. In this case log likelihood ratios are linear functions of the observation.

**Assumption 4.** *Every observation probability distribution $\mathbb{P}_\theta[\cdot|a]$, for $a \in \mathcal{A}$ and $\theta \in \Theta^{(1)} \cup \{\theta_0\}$, has either a density[4] or a mass function, denoted by $f_\theta(\cdot|a)$. Moreover, there exists $r > 0$ such that for any $\theta, \theta', \theta'' \in \Theta^{(1)} \cup \{\theta_0\}$, and $a \in \mathcal{A}$, the log likelihood ratio $\frac{f_{\theta'}(X|a)}{f_{\theta''}(X|a)}$ is $r$-subgaussian [5] under $X \sim \mathbb{P}_\theta[\cdot|a]$.*

---

[4] with respect to a standard reference measure, e.g., Lebesgue measure

[5] A random variable $X$ is $r$-subgaussian under $X \sim \mathbb{P}$ if $\forall \lambda \in \mathbb{R}$: $\mathbb{E}[e^{\lambda(X - \mathbb{E}[X])}] \leq e^{\lambda^2 r/2}$.

This assumption is common in the statistical inference literature; we use it to be able to control the fluctuations of the exploration and exploitation CUSUM statistics in the algorithm via standard subgaussian concentration tools. A concrete example of a setting that meets Assumptions 1-4 is the linear observation model described in Section 3 and Fig. 1. We are now in a position to state our key theoretical result.

**Theorem 3** (False alarm and detection delay for general change point). *Under assumptions 1-4, the following conclusions hold for $\epsilon$-GCD (Algorithm 1) run with the log-likelihood observation function $g_\theta(x, a) \equiv \log \frac{f_\theta(x|a)}{f_{\theta_0}(x|a)}$.*

*1. (Time to false alarm) Let $\alpha \in (0, 1)$ and $m \in \mathbb{N}$. If the stopping threshold $\beta$ is set as $\beta \geq \frac{m|\Theta^{(1)}|}{\alpha}$, then the stopping time satisfies $\mathbb{P}^{(\infty)}[\tau < m] \leq \alpha$.*

*2. (Detection delay) For a change from $\theta_0$ to $\theta^* \in \Theta^{(1)}$ occurring at time $\nu \in \mathbb{N}$,*

$$\mathbb{E}^{(\nu,\theta^*)}\left[(\tau - \nu)^+\right] \leq \frac{8 \log \beta}{(1-\epsilon)\max_a D\left(\theta^*(a)||\theta^{(0)}(a)\right)} + O\left(poly\left(\frac{1}{\epsilon}\right), \mathbb{E}^{(\infty)}\left[Q_\nu^{(1)}\right], \mathcal{P}\right),$$

(1)

*provided $\pi(a_{\theta^*}^*) > 0$, with $Q_\nu^{(1)} := \left(Q_\nu^{(1)}(\theta)\right)_{\theta \in \Theta^{(1)}}$ denoting the explore queue statistics for all parameters and $\mathcal{P} \equiv (\mathbb{P}_\theta[\cdot|a])_{\theta \in \{\theta_0\} \cup \Theta^{(1)}}$ all the observation distributions.*

*Remark.* We note that $\beta$ is an internal, algorithmic parameter, whose setting is spelt out in order that the $(m, \alpha)$ false alarm constraint can be met. Thus, the first part of Theorem 3 is a 'correctness' statement[6]

**Interpretation of the Detection Delay Bound.** The first term in the detection delay bound (1) can be interpreted as the time for an *oracle* fixed-action strategy (e.g., CUSUM), that always plays the most informative action for $\theta^*$, to stop. The second term, on the other hand, is a bound on the time taken by the algorithm to *learn* to play the most informative action for $\theta^*$ (details follow in proof sketch). We omit the precise dependence of the second term on the problem structure here for brevity, but detail it explicitly in the appendix. Specifically, for small forced exploration rates $\epsilon$, the second term depends on the information geometry of the problem approximately as $\frac{1}{\epsilon^4}\sum_{\theta \in \Theta^{(1)}:a_\theta^* \neq a_{\theta^*}^*} \frac{1}{\Delta_\theta^4}$. Here, for each candidate post-change parameter $\theta \in \Theta^{(1)}$, its 'gap' $\Delta_\theta$ is a measure of how difficult it is for the algorithm to eliminate $\theta$ as an estimate for the true post-change parameter $\theta^*$ during this parameter-learning phase. We formally define it as $\Delta_\theta := \bar{D}_{\theta^*,\theta_0} - \frac{1}{2}(\bar{D}_{\theta^*,\theta_0} + (\bar{D}_{\theta^*,\theta_0} - \bar{D}_{\theta^*,\theta})^+)$, where $\bar{D}_{\theta^*,\theta} := \sum_{a \in \mathcal{A}} \pi(a) D(\theta^*(a)||\theta(a))$ is the average information divergence between parameters $\theta^*, \theta$ offered by playing from the exploration distribution $\pi$. With finer analysis, the dependence on gaps and $\epsilon$ could be improved. We omit it for simplicity.

We now turn to the (additive & linear; see the appendix) dependence on $\mathbb{E}^{(\infty)}\left[Q_\nu^{(1)}\right]$ of the second term. The insight as to why this arises is as follows. The 'learning' phase for $\theta^*$, after a change takes place at time $\nu$, can be seen as a 'race' between competing log-likelihood queues of all the parameters vying to become the MLE $\hat{\theta}_t$. At the change time, each non-ground truth queue $Q_\nu^{(1)}(\theta)$ is, on average, at level $\mathbb{E}^{(\infty)}\left[Q_\nu^{(1)}(\theta)\right]$, representing its initial 'advantage' over the ground-truth queue $Q_\nu^{(1)}(\theta^*)$ which in the worst case could be at level 0. Thus, the difference $\mathbb{E}^{(\infty)}\left[Q_\nu^{(1)}(\theta)\right] - Q_\nu^{(1)}(\theta^*) \leq Q_\nu^{(1)}(\theta)$ is the extra amount of 'time work' that the $\theta^*$ queue must do to overcome the $\theta$ queue. A special case of the result is for $\nu = 1$ in which case all queues start out at 0. For general $\nu$, one can establish that $\mathbb{E}^{(\infty)}\left[Q_\nu^{(1)}(\theta)\right]$ must be finite, by noticing that each queue is non-negative and accumulates arrivals with negative mean before the change, and applying standard queueing stability arguments. This also indicates that the detection delay bound is roughly invariant given the location of the change point $\nu$.

**Optimality at low false-alarms.** The result implies that in the 'small' time to false alarm regime, i.e., $\alpha \to 0$, $m = \omega(\log(1/\alpha))$ and $\log(m) = o(\log(\alpha))$, the detection delay when $\beta$ is set as

---

[6]This is, at a high level, akin to specifying a PAC algorithm's correctness requirement and measuring its sample complexity on the other hand, i.e., insist that its returned answer be $\epsilon$-correct with probability at least $1 - \delta$, and then analyze how long it takes to learn and return an answer. The former corresponds to the false alarm constraint and the latter to measuring detection delay here.

above is dominated by the first term: $\mathbb{E}^{(\nu,\theta^*)}[\tau] = O\left(\frac{\log\beta}{(1-\epsilon)\max_a D(\theta^*(a)||\theta^{(0)}(a))}\right)$. This matches, order-wise, the universal lower bound of Theorem 2 up to the algorithm-dependent multiplicative factor $1/(1-\epsilon)$, which may be interpreted as a 'penalty' for the forced exploration mechanism.

**Benefits of Adaptivity.** Let us consider a concrete setting to elucidate the detection delay bound. In this example, there are $d$ sensing actions, represented by the canonical basis vectors in $\mathbb{R}^d$. The pre-change parameter is $0 \in \mathbb{R}^d$, and there are $d$ candidate post-change parameters, each of which is a canonical basis vector $e_i$, $i \in [d]$, multiplied by a constant $\delta$. The observation from applying action $a$ at system parameter $\theta$ is Gaussian distributed with mean $\langle a, \theta \rangle$ and variance 1. Thus, the aim is to detect a (sparse) change of magnitude $\delta$ in one of the $d$ coordinates in $\theta_0 = 0$, when at any time only a single coordinate can be noisily sensed. Suppose $\theta^* = \delta e_1$ without loss of generality, for which the most informative action (in fact, the only informative one) is $a_{\theta^*}^* = e_1$. The first term in the detection delay given by Theorem 3 is then $O\left(\frac{\log(1/\alpha)}{(1-\epsilon)\delta^2}\right)$. This is a factor of $d$ smaller than $\Omega\left(d \cdot \frac{\log(1/\alpha)}{(1-\epsilon)\delta^2}\right)$ that a standard CUSUM rule with non-adaptive uniform sampling over coordinates would achieve – this is seen by applying the lower bound (Theorem 2) to the case of a single (trivial action) where the divergence in the denominator reduces to the *average* divergence across all coordinates: $O(\delta^2/d)$. For estimating the second term, we calculate that for each $\theta = \delta e_i$ with $i \neq 1$, $\Delta_\theta = \bar{D}_{\theta^*,\theta_0} - \frac{(\bar{D}_{\theta^*,\theta_0} + (\bar{D}_{\theta^*,\theta_0} - \bar{D}_{\theta^*,\theta})^+)}{2} = \frac{\delta^2}{2d} - \frac{\frac{\delta^2}{2d} - (\frac{\delta^2}{2d} - \frac{\delta^2}{d})^+}{2} = \frac{\delta^2}{2d} - \frac{\frac{\delta^2}{2d} - 0}{2} = \frac{\delta^2}{4d}$. Thus, the second term representing the time to learn the optimal sensing action scales as $O\left(\frac{(d-1)d^4}{\epsilon^4\delta^8}\right) = O\left(\frac{d^5}{\epsilon^4\delta^8}\right)$. Although we have not optimized dependence on $d, \epsilon$ and $\delta$, the overall bound still gives an idea of how the active sensing problem gets harder as the dimension $d$ grows or as the minimum change amount $\delta$ changes, making it akin to finding a 'needle in a haystack'.

**Sketch of the Proof of Theorem 3.** This section lays down the key arguments involved in the proof of Theorem 3 for bounding the false alarm time and detection delay of the $\epsilon$-GCD algorithm.

**1. Bounding the probability of early stopping under no change.** The stopping time $\tau$ for $\epsilon$-GCD, by line 4 in Algorithm 1, is equivalent to the first instant $t$ when the worst-case (over $\Theta^{(1)}$) CUSUM statistic computed over exploitation data, i.e., $\max_{\theta \in \Theta^{(1)}} Q_t^{(2)}(\theta) \equiv \max_{\theta \in \Theta^{(1)}} \max_{1 \leq s \leq t} \prod_{\ell=s}^{t-1}\left(\frac{f_\theta(X_\ell|A_\ell)}{f_{\theta_0}(X_\ell|A_\ell)}\right)^{(1-E_\ell)}$ exceeds the level $\beta \geq 1$. Under no change, each observation $X_\ell$ is distributed as $\mathbb{P}_{\theta_0}[\cdot|A_\ell]$ given $A_\ell$, and the product $\prod_{\ell=s}^{t-1}\left(\frac{f_\theta(X_\ell|A_\ell)}{f_{\theta_0}(X_\ell|A_\ell)}\right)^{(1-E_\ell)}$, over $t = s, s+1, s+2, \ldots$, behaves as a (standard) likelihood ratio martingale with unit expectation under the appropriate filtration. Hence, the chance that the largest among this ensemble of martingales, one for each $\theta \in \Theta^{(1)}$ and starting time $s \in [m]$, rises above $\beta$ before time $m$ is bounded using a union bound and Doob's inequality for each individual martingale, yielding the probability bound $\frac{m|\Theta^{(1)}|}{\beta}$.

**2. Control of the detection delay.** Suppose that the change takes place at time $\nu = 1$ to the post-change parameter $\theta^* \in \Theta^{(1)}$. The proof strategy is to show, with high probability, that $\tau$ is no more than $t_1 + t_2$, where

• $t_1$ is an upper bound for the time taken for the plug-in estimate $\hat{\theta}_t$, of the post-change parameter, to 'settle' to $\theta^*$. In other words, we show that after time $t_1$ it is very unlikely that an action other than $a_{\theta^*}^* = \arg\max_{a \in \mathcal{A}} D(\theta^{(0)}(a)||\theta^*(a))$ is played in every exploitation round.

• $t_2$ is an upper bound for the time taken for the worst-case CUSUM statistic $\max_{\theta \in \Theta^{(1)}} Q_t^{(2)}(\theta)$ to grow to the level $\beta$ *assuming that the optimal (i.e., most informative) action $a_{\theta^*}^*$ is always played at all exploitation rounds*.

For clarity of exposition, we will assume that we are in the setting of *linear measurements* in $\mathbb{R}^d$ with *additive standard Gaussian noise* – this makes KL divergences easy to interpret as Euclidean distances – and that the changepoint is at $\nu = 1$.

**Step 1: Finding $t_1$ (time to learn $\theta^*$).** We observe that the MLE $\hat{\theta}_t$ at time $t$ can be written as the parameter $\theta$ associated with the largest stochastic process $\max_{v=1}^{t-1} \log J_{v,t-1}(\theta)$, one for each $\theta$:
$$\hat{\theta}_t = \arg\max_{\theta \in \Theta^{(1)}} \max_{v=1}^{t-1} \log J_{v,t-1}(\theta), \text{ where } J_{v,t-1}(\theta) := \prod_{\ell=v}^{t-1}\left(\frac{f_\theta(X_\ell|A_\ell)}{f_{\theta_0}(X_\ell|A_\ell)}\right)^{E_\ell}.$$

Under the post-change distribution $\mathbb{P}^{(1,\theta^*)}$, $\log J_{v,t-1}(\theta)$ can be seen to evolve (as a function of $t$) as a random walk with drift $\epsilon(\|\theta^* - \theta\|_H^2 - \|\theta\|_H^2)$. Here, $\|\cdot\|_H$ is the usual matrix-weighted Euclidean norm in $\mathbb{R}^d$, governed by how much different directions are explored in expectation[7] by the exploration distribution $\pi$: $H = \sum_{a \in \mathcal{A}} \pi(a) a a^T$. Note that $\|\theta^* - \theta\|_H^2$ here is the KL divergence between the distributions of the observation that results when an action sampled from $\pi$ is played, under parameters $\theta^*$ and $\theta$.

The preceding discussion implies that the drift is the largest (and positive) for the random walk corresponding to the true post-change parameter $\theta = \theta^*$. The remainder of this part of the proof uses martingale concentration tools (subgaussian Chernoff and maximal Hoeffding bounds) to find the time $t_1$ at which the fastest-growing random walk, $\log J_{v,t}(\theta^*)$ dominates all other 'competing' random walks $\log J_{v,t}(\theta)$, $\theta \neq \theta^*$.

**Step 2: Finding $t_2$ (time to stop under optimal action plays).** In a manner similar to that of Step 1, we observe that the logarithm of the log-likelihood ratio for $\theta^*$ w.r.t. $\theta_0$ computed over only *exploitation* rounds, i.e., $\log \prod_{\ell=v}^{t-1} \left( \frac{f_\theta(X_\ell|A_\ell)}{f_{\theta_0}(X_\ell|A_\ell)} \right)^{1-E_\ell}$, evolves as a random walk with drift rate $(1-\epsilon)\|\theta^*\|_H$. A Chernoff bound can then be used to control the probability of the 'bad' event that this growing random walk has *not* crossed the level $\log \beta$ in a certain time duration $t_2$ ($t_2$ does not appear explicitly in the main derivation).

# 6   Experiments

Our goal in this section is to illustrate various aspects of our theory through experiments on synthetic environments, and to explore the performance of the proposed $\epsilon$-GCD algorithm on a setting based on a real-world dataset. All experiments were performed on a laptop with Intel Core i5 CPU and 8GB of RAM, and take under an hour to execute. The closely related prior works on this topic [ZM20; Gun+21] are more specific and are not compatible with many of our structured

| Size | Oracle | $\epsilon$-GCD+ | $\epsilon$-GCD | URS |
|------|--------|----------------|----------------|-----|
| 10 | $30 \pm 5$ | $98 \pm 62$ | $112 \pm 74$ | $306 \pm 76$ |
| 15 | $31 \pm 5$ | $129 \pm 95$ | $158 \pm 119$ | $460 \pm 115$ |
| 20 | $31 \pm 5$ | $163 \pm 128$ | $196 \pm 156$ | $616 \pm 153$ |
| 25 | $31 \pm 5$ | $191 \pm 154$ | $253 \pm 216$ | $764 \pm 194$ |

Table 1: Observed mean and standard deviation for the simulated stopped time for varying graph-sizes with pointy action set $\mathcal{A}_1$, for change occurring at $\nu = 40$.

scenarios (see Sec. 2 and Appendix E). As alternatives, we propose several natural baselines and compare against our method to benchmark performance.

**Uniform Random Sampling (URS).** At every round we sample each action in $\mathcal{A}$ uniformly at random. Our goal with URS is to calibrate adaptation gains.

**$\epsilon$-GCD (Ours).** This is our proposed scheme. A point to be noted here (as seen from Algorithm 1 is that the data used for making a stopping decision ("exploitation") and that for estimating the changed distribution ("exploration") are non-overlapping, and as such leads to inefficient data usage.

**$\epsilon$-GCD+.** To reduce the aforementioned data inefficiency, we consider the variant, the parameter estimation part also utilizes both data collected during exploration rounds and the data collected for the exploitation rounds.

**Oracle.** Here we consider an omniscient strategy in that the algorithm is aware of both the pre-change and post-change distributions ahead of time, and consequently does not require exploration to estimate the post-change parameter. On the other hand the method has no knowledge of stopping time, and thus the problem reduces to the conventional quickest change detection. The Oracle characterizes the gold-standard in that no adaptation is required, and it gives the absolute smallest detection delay for any BQCD problem.

**Synthetic Experiments.** We conduct synthetic experiments for the model introduced in Section 3 under the bold item titled "illustrative example." We explore how average detection delay varies under controlled variation of various parameters such as changepoint location, dimensionality and structure of the ambient space, and the type of action sets. Our objective is twofold: (a) Illustrate gains due to adaptivity of proposed $\epsilon$-GCD over the non-adaptive method, where actions are chosen uniformly at random; (b) Demonstrate "near" optimality by baselining against Oracle.

---

[7]This matrix also arises commonly in linear design of experiments as the Kiefer-Wolfowitz matrix [LS20].

We report results for the case when the ambient space is a line graph (see Illustrative example in Sec. 3). Nodes are interpreted as physical locations, and take values in $[N]$. Nodes $j, k$ are connected if $|j - k| = 1$. Each node $n \in [N]$ offers a Gaussian-distributed observation depending on the changepoint $\nu \in \mathbb{N}$.

*Isolated and Structured Anomalies.* The vector of change parameters $\theta = [\theta_n]$ are of two types: (a) Isolated singleton change, namely, $\theta_n \in \{0, 1\}$ and $\sum_{n \in [N]} \theta_n = 1$; (b) Structural changes, i.e., $\text{Supp}(\theta) = \{n \in [N] : \theta_n \neq 0\}$ is k-connected set with $\theta \in \{0, 1\}^N$.

*Diffuse and 'Pointy' Action sets.* In a parallel fashion we consider two types of action sets. The set $\mathcal{A}_1$, is pointy, namely, $|\text{Supp}(\mathcal{A}_1)| = 1$, which allows probing only single nodes. The set $\mathcal{A}_2$ is diffuse where only a connected subset of nodes can be queried. In either case, the observation received on an action is as described in Sec. 3.

**Key Findings.** We report results for pointy actions and isolated anomalies, and defer other settings to the appendix. We choose $\sigma^2 = 0.5$, $\beta$ such that the false alarms are about $1\%$ for the Oracle, and a forced exploration rate $\epsilon = 0.2$. Our results are averaged over 5000 Monte-Carlo runs. With this choice for $\beta$ we did not observe false alarms in any of the algorithms.

*Gains from Adaptivity.* The fact that we perform substantially better than URS uniformly across all of the experiments demonstrates our gains through adaptive sampling.

*Data Inefficiency.* The fact that the proposed method closely tracks $\epsilon$-GCD+ suggests that our decoupled scheme is capable of fully leveraging the observations, and the cost of decoupled exploration is small.

*Optimality.* The fact that the proposed method tracks Oracle qualitatively across all experiment setups indicates that our method is close to optimal. This is because through adaptivity we

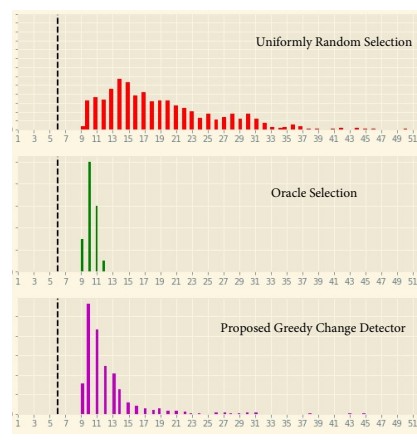

Figure 2: Audio based change detection of machine anomaly: Histogram of stopping times by URS, Oracle, and $\epsilon$-GCD. Histograms for Oracle and $\epsilon$-GCD demonstrates clear adaptivity gains.

achieve performance similar to a competitor who requires no adaptation (recall that oracle knows the change parameters and thus does not need to explore). In point of fact we note in passing that any other method would at best lie between our proposed method and the Oracle.

*Variation with Changepoint $\nu$.* For a fixed graph size, we observed that the average expected delay is relatively constant for all methods, which is consistent with Theorem 3.

**Audio based recognition of machine anomalies.** We experiment using the MIMII audio dataset [Pur+19b]. Detailed specifics are in the appendix. The dataset has four machines (ID00,ID02,ID04,ID06), each equipped with audio sensors recording the health of the machine. There are three types of anomaly (rail damage, loose belt, no grease) which can occur at any time, in any machine. Corresponding to each anomaly there is an audio stream, and the anomaly occurs in one of the four machines at an arbitrary time. For each machine, the dataset contains audio-streams of 1000 normal and 300 abnormal files, and each audio-stream is about 10 seconds long.

*Audio Processing.* For each audio-stream we train autoencoders on normal data using mel-spectrogram features, and fit Gaussians to the reconstruction errors. This results in pre- and post-change parameters for each machine's autoencoded reconstruction error score, corresponding to normal and abnormal operation.

*Experiment.* To simulate BQCD, we introduced anomalies in machine bearing ID00 as follows. We concatenated 6 normal files and 54 abnormal files chosen uniformly at random from machine ID00. For the other machines we concatenated 60 normal files at random. The 60 files correspond to 600 seconds. Our changepoint corresponds to 6th file, which we denote as $\nu = 6$ and our task is to detect this change. Note that the machine ID and the changepoint are both not known to the learner. Our results are depicted as histograms for changepoints of anomaly detection in Fig. 2. As observed, we notice that $\epsilon$-GCD's performance is close to Oracle both in mean and distribution, while URS exhibits larger delay and significant variance. Appendix D presents histograms for a larger changepoint.

