## Acknowledgments and Disclosure of Funding

A. G. and B. L. were supported by the Aerospace Network Research Consortium (ANRC) Grant on Airplane IOT Data Management. V. S. was supported by the Army Research Office Grant W911NF2110246, and National Science Foundation grants CCF-2007350 and CCF-1955981. The authors are grateful to Rajesh Sundaresan, Himanshu Tyagi and Manjunath Krishnapur for helpful discussions, and to the anonymous program committee members for their valuable feedback.

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

# Appendix

## A  Proof of Theorem 2

We in fact establish the following more general result.

**Theorem 4.** *Let* $0 < \alpha \leq \frac{1}{10}$, $m \geq 1$ *and* $\nu \in \mathbb{N}$. *If a bandit changepoint algorithm satisfies* $\mathbb{P}^{(\infty)} \left[ \tau < \nu + m \mid \nu \leq \tau \right] \leq \alpha$, *then for any* $\theta^* \in \Theta^{(1)}$,

$$\mathbb{E}^{(\nu, \theta^*)} \left[ \tau - \nu \mid \tau \geq \nu \right] \geq \min \left\{ \frac{\frac{1}{20} \log \frac{1}{\alpha}}{\max_{a \in \mathcal{A}} D\left( \theta^*(a) || \theta_0(a) \right)}, \frac{m}{2} \right\}.$$

Note that Theorem 2 is the special case[8] of $\nu = 1$.

We first recall and/or put down some preliminaries before embarking upon the proof.

**Definition: Problem instance.** For any changepoint time $\nu \in \{1, 2, \ldots\} \cup \{\infty\}$ and post-change parameter $\theta \in \Theta^{(1)}$, we call the pair $(\nu, \theta)$ an *instance* of the bandit changepoint detection problem. Note that if $\nu = \infty$, then it is immaterial what the post-change parameter $\theta$ is, since there is effectively no change in the distribution; thus we will omit $\theta$, or write $(\infty, *)$, if $\nu = \infty$ for ease of notation. The instance, along with the sampling algorithm and (known) $\theta^{(0)}$, completely determines the distribution of trajectories generated by the operation of the algorithm.

**Bandit changepoint detection algorithm:** Recall the definition of an algorithm for the bandit changepoint detection problem: It is a rule that maps the history $I_t$ of actions and observations to (1) a decision $U_t \in \{0, 1\}$ to stop playing actions, and (2) if not stopping ($U_t = 0$), then an action $A_t$ to play in round $t$. Here,

$$I_t = (V_0, U_0, A_1, X_1, V_1, U_1, A_2, X_2, V_2, \ldots, V_{t-2}, U_{t-2}, A_{t-1}, X_{t-1}, V_t),$$

where at any time instant $s \geq 0$, $V_s$ represents independent, internal randomness available to the algorithm, $U_s$ is an indicator random variable for the event that the algorithm decides to stop playing before taking the $(s + 1)$-st action (i.e., after playing $s$ actions), and $X_s$ is the observation from playing arm $A_s$ in round $s$.

*Proof of Theorem 4.* We first establish an auxiliary lemma about the explicit form for the divergence of the conditional distribution of a trajectory.

Let $\text{Law}_E^{(i, \theta)}(I_\tau)$ denote the probability distribution of the random trajectory $I_\tau$ conditioned on the event $E$, when the algorithm is run on the instance $(i, \theta)$.

**Lemma 1.** *For any parameter* $\theta \in \Theta^{(1)}$, $\eta \in \mathbb{N}$ *and* $E \in \sigma(I_\eta)$,

$$D\left( \text{Law}_E^{(\eta, \theta)}(I_\tau) || \text{Law}_E^{(\infty)}(I_\tau) \right) = \sum_{a \in \mathcal{A}} D\left( \theta(a) || \theta_0(a) \right) \mathbb{E}^{(\eta, \theta)} \left[ \left( N_\tau(a) - N_{\eta-1}(a) \right)^+ \mid E \right].$$

*Proof.* For the sake of convenience we show the argument assuming that the trajectory $I_\tau$ is a discrete random object, i.e., it has a probability mass function. (It is straightforward, but notationally heavier, to extend it to the case of general measures by using Radon-Nikodym derivatives.)

---

[8]We assume that $\tau \geq 1$ with probability 1.

We have

$$D\left(\mathrm{Law}_E^{(\eta,\theta)}(I_\tau)\|\mathrm{Law}_E^{(\infty)}(I_\tau)\right)$$

$$= \sum_{\omega\equiv(v_0,u_0,a_1,x_1,v_1,\ldots,v_{\tau-1},u_\tau=1)\in E} \mathbb{P}^{(\eta,\theta)}\left[I_\tau=\omega\mid E\right]\log\frac{\mathbb{P}^{(\eta,\theta)}\left[I_\tau=\omega\mid E\right]}{\mathbb{P}^{(\infty)}\left[I_\tau=\omega|E\right]}$$

$$\overset{(a)}{=} \sum_{\omega\equiv(v_0,u_0,a_1,x_1,v_1,\ldots,v_{\tau-1},u_\tau=1)\in E} \mathbb{P}^{(\eta,\theta)}\left[I_\tau=\omega\mid E\right]\log\frac{\mathbb{P}^{(\eta,\theta)}\left[I_\tau=\omega\right]}{\mathbb{P}^{(\infty)}\left[I_\tau=\omega\right]}$$

$$= \sum_{\omega\in E} \mathbb{P}^{(\eta,\theta)}\left[I_\tau=\omega\mid E\right]\log\frac{\mathbb{P}^{(\eta,\theta)}\left[v_0\right]\mathbb{P}^{(\eta,\theta)}\left[u_0|v_0\right]\mathbb{P}^{(\eta,\theta)}\left[a_1|u_0,v_0\right]\mathbb{P}^{(\eta,\theta)}\left[x_1|a_1\right]\cdots}{\mathbb{P}^{(\infty)}\left[v_0\right]\mathbb{P}^{(\infty)}\left[u_0|v_0\right]\mathbb{P}^{(\infty)}\left[a_1|u_0,v_0\right]\mathbb{P}^{(\infty)}\left[x_1|a_1\right]\cdots}$$

$$\overset{(b)}{=} \sum_{\omega\in E} \mathbb{P}^{(\eta,\theta)}\left[I_\tau=\omega\mid E\right]\log\frac{\mathbb{P}^{(\eta,\theta)}\left[x_1|a_1\right]\mathbb{P}^{(\eta,\theta)}\left[x_2|a_2\right]\cdots}{\mathbb{P}^{(\infty)}\left[x_1|a_1\right]\mathbb{P}^{(\infty)}\left[x_2|a_2\right]\cdots}$$

$$= \sum_{\omega\in E} \mathbb{P}^{(\eta,\theta)}\left[I_\tau=\omega\mid E\right]\sum_{t=1}^{\tau-1}\log\frac{\mathbb{P}^{(\eta,\theta)}\left[x_t(\omega)|a_t(\omega)\right]}{\mathbb{P}^{(\infty)}\left[x_t(\omega)|a_t(\omega)\right]}$$

$$\overset{(c)}{=} \frac{1}{\mathbb{P}^{(\infty)}\left[E\right]}\mathbb{E}^{(\eta,\theta)}\left[\sum_{t=1}^{\tau-1}\log\frac{\mathbb{P}^{(\eta,\theta)}\left[X_t|A_t\right]}{\mathbb{P}^{(\infty)}\left[X_t|A_t\right]}\mathbf{1}\left\{E\right\}\right].$$

Here, $(a)$ and $(c)$ both follow because $\mathbb{P}^{(\eta,\theta)}\left[E\right]=\mathbb{P}^{(\infty)}\left[E\right]$ due to $E\in\sigma(I_\eta)$, and $(b)$ is because the algorithm's decisions and internal randomness $v_0,u_0,a_1,a_2$, etc. do not depend on the probability distribution of the environment generating the observations. We continue further, writing

$$\mathbb{P}^{(\infty)}\left[E\right]\cdot D\left(\mathrm{Law}_E^{(\eta,\theta)}(I_\tau)\|\mathrm{Law}_E^{(\infty)}(I_\tau)\right)$$

$$= \mathbb{E}^{(\eta,\theta)}\left[\mathbf{1}\left\{E\right\}\sum_{t=1}^{(\eta-1)\wedge(\tau-1)}\log\frac{\mathbb{P}^{(\eta,\theta)}\left[X_t|A_t\right]}{\mathbb{P}^{(\infty)}\left[X_t|A_t\right]}+\mathbf{1}\left\{E\right\}\sum_{t=(\eta-1)\wedge(\tau-1)+1}^{\tau-1}\log\frac{\mathbb{P}^{(\eta,\theta)}\left[X_t|A_t\right]}{\mathbb{P}^{(\infty)}\left[X_t|A_t\right]}\right]$$

$$= \mathbb{E}^{(\eta,\theta)}\left[\mathbf{1}\left\{E\right\}\sum_{t=\eta\wedge\tau}^{\tau-1}\log\frac{\mathbb{P}^{(\nu,\theta)}\left[X_t|A_t\right]}{\mathbb{P}^{(\infty)}\left[X_t|A_t\right]}\sum_{a\in\mathcal{A}}\mathbf{1}\left\{A_t=a\right\}\right],$$

because $E\in\sigma(I_\eta)$. Thus,

$$\mathbb{P}^{(\infty)}\left[E\right]\cdot D\left(\mathrm{Law}_E^{(\eta,\theta)}(I_\tau)\|\mathrm{Law}_E^{(\infty)}(I_\tau)\right)$$

$$= \sum_{a\in\mathcal{A}}\sum_{t=\eta\wedge\tau}^{\tau-1}\mathbb{E}^{(\eta,\theta)}\left[\log\frac{\mathbb{P}^{(\nu,\theta)}\left[X_t|A_t\right]}{\mathbb{P}^{(\infty)}\left[X_t|A_t\right]}\mathbf{1}\left\{A_t=a,E\right\}\right]$$

$$= \sum_{a\in\mathcal{A}}\sum_{t=\eta\wedge\tau}^{\tau-1}\mathbb{E}^{(\eta,\theta)}\left[\mathbf{1}\left\{A_t=a,E\right\}\mathbb{E}^{(\eta,\theta)}\left[\log\frac{\mathbb{P}^{(\eta,\theta)}\left[X_t|A_t\right]}{\mathbb{P}^{(\infty)}\left[X_t|A_t\right]}\Big|\mathbf{1}\left\{A_t=a,E\right\}\right]\right]$$

$$\overset{(d)}{=} \sum_{a\in\mathcal{A}}\sum_{t=\eta\wedge\tau}^{\tau-1}\mathbb{E}^{(\eta,\theta)}\left[\mathbf{1}\left\{A_t=a,E\right\}D\left(\theta(a)\|\theta_0(a)\right)\right]$$

$$= \sum_{a\in\mathcal{A}}D\left(\theta(a)\|\theta_0(a)\right)\mathbb{E}^{(\eta,\theta)}\left[\mathbf{1}\left\{E\right\}\left(N_\tau(a)-N_{\eta-1}(a)\right)^+\right],$$

where $(d)$ is due to $E\in\sigma(I_\eta)$ and the Markov property of the algorithm's trajectory. This completes the proof. □

Returning to the proof of the theorem, we split the analysis into two cases depending on the value of the conditional probability $\mathbb{P}^{(\nu,\theta^*)}\left[\tau<\nu+m\mid\nu\leq\tau\right]$ of stopping before an additional time $m$ after having crossed the actual changepoint $\nu$.

**Case 1:** $\mathbb{P}^{(\nu,\theta^*)}\left[\tau < \nu + m \mid \nu \leq \tau\right] \geq \frac{1}{2}$. In this case, applying the data processing inequality for KL divergence [Cov99] to the two (conditional) input distributions $\mathrm{Law}_{\nu \leq \tau}^{(\nu,\theta^*)}(I_\tau) \equiv \mathbb{P}^{(\nu,\theta^*)}\left[I_\tau \in \cdot \mid \nu \leq \tau\right]$ and $\mathrm{Law}_{\nu \leq \tau}^{(\infty)}(I_\tau) \equiv \mathbb{P}^{(\infty)}\left[I_\tau \in \cdot \mid \nu \leq \tau\right]$, with the data processing function $f(I_\tau) := \mathbf{1}\{\nu \leq \tau < \nu + m\}$, yields

$$D\left(\mathrm{Law}_{\nu \leq \tau}^{(\nu,\theta^*)}(I_\tau)||\mathrm{Law}_{\nu \leq \tau}^{(\infty)}(I_\tau)\right)$$
$$\geq D\left(\mathrm{Ber}\left(\mathbb{P}^{(\nu,\theta^*)}\left[\tau < \nu + m \mid \nu \leq \tau\right]\right)||\mathrm{Ber}\left(\mathbb{P}^{(\infty)}\left[\tau < \nu + m \mid \nu \leq \tau\right]\right)\right).$$

Together with Lemma 1 for the event $E := \{\nu \leq \tau\}$, this gives

$$\sum_{a \in \mathcal{A}} D\left(\theta^*(a)||\theta_0(a)\right)\mathbb{E}^{(\nu,\theta^*)}\left[(N_\tau(a) - N_{\eta-1}(a))^+ \mid \nu \leq \tau\right]$$
$$\geq D\left(\mathrm{Ber}\left(\mathbb{P}^{(\nu,\theta^*)}\left[\tau < \nu + m \mid \nu \leq \tau\right]\right)||\mathrm{Ber}\left(\mathbb{P}^{(\infty)}\left[\tau < \nu + m \mid \nu \leq \tau\right]\right)\right)$$
$$\overset{(a)}{\geq} \mathbb{P}^{(\nu,\theta^*)}[\tau < \nu + m \mid \nu \leq \tau]\log\frac{\mathbb{P}^{(\nu,\theta^*)}[\tau < \nu + m \mid \nu \leq \tau]}{\mathbb{P}^{(\infty)}[\tau < \nu + m \mid \nu \leq \tau]} - \ln 2$$
$$\overset{(b)}{\geq} \frac{1}{2}\log\frac{1/2}{\alpha} - \ln 2 \geq \frac{1}{20}\log\frac{1}{\alpha},$$

by Lemma 2, where $(a)$ is due to [GMS19] and $(b)$ is by hypothesis. We now divide both sides by $\sum_{a \in \mathcal{A}}\mathbb{E}^{(\nu,\theta^*)}\left[(N_\tau(a) - N_{\eta-1}(a))^+ \mid \nu \leq \tau\right] = \mathbb{E}^{(\nu,\theta^*)}\left[(\tau - \nu)^+ \mid \nu \leq \tau\right]$, and use the fact that the maximum is at least a convex combination, to get

$$\max_{a \in \mathcal{A}} D\left(\theta^*(a)||\theta_0(a)\right) \geq \frac{1}{20}\frac{\log(1/\alpha)}{\mathbb{E}^{(\nu,\theta^*)}\left[(\tau - \nu)^+ \mid \nu \leq \tau\right]},$$

giving one part of the theorem.

**Case 2:** $\mathbb{P}^{(\nu,\theta^*)}\left[\tau < \nu + m \mid \nu \leq \tau\right] < \frac{1}{2}$. In this case, we have

$$\mathbb{E}^{(\nu,\theta^*)}\left[(\tau - \nu)^+ \mid \nu \leq \tau\right] \geq m \cdot \mathbb{P}^{(\nu,\theta^*)}\left[\tau \geq \nu + m \mid \nu \leq \tau\right] \geq \frac{m}{2},$$

giving the other part of the theorem.

**Lemma 2.** *For $0 < x < \frac{1}{10}$, we have $\frac{1}{2}\log\frac{1}{2x} - \log 2 \geq \frac{1}{20}\log\frac{1}{x}$.*

*Proof.* The proof is by basic calculus. $\qquad\square$

$\square$

## B  Proof of Theorem 3

### B.1  Time to false alarm

The $\epsilon$-GCD algorithm stops at the first time $t$ when the largest exponentiated 'exploitation queue' CUSUM statistic, i.e.,

$$\max_{\theta \in \Theta^{(1)}} \exp\left(Q_t^{(2)}(\theta)\right) := \max_{\theta \in \Theta^{(1)}}\max_{1 \leq s \leq t}\prod_{\ell=s}^{t-1}\left(\frac{f_\theta(X_\ell|A_\ell)}{f_{\theta_0}(X_\ell|A_\ell)}\right)^{(1-E_\ell)},$$

exceeds $\beta$.

For $m \in \mathbb{N}$, let us compute

$$\mathbb{P}^{(\infty)}\left[\tau < m\right] = \mathbb{P}^{(\infty)}\left[\exists t < m: \max_{\theta \in \Theta^{(1)}} \max_{1 \leq s \leq t} \prod_{\ell=s}^{t-1} \left(\frac{f_\theta(X_\ell|A_\ell)}{f_{\theta_0}(X_\ell|A_\ell)}\right)^{(1-E_\ell)} \geq \beta\right]$$

$$\leq \mathbb{P}^{(\infty)}\left[\exists \theta \in \Theta^{(1)}, s \in [m]: \max_{s \leq t \leq m} \prod_{\ell=s}^{t-1} \left(\frac{f_\theta(X_\ell|A_\ell)}{f_{\theta_0}(X_\ell|A_\ell)}\right)^{(1-E_\ell)} \geq \beta\right]$$

$$\leq \sum_{\theta \in \Theta^{(1)}} \sum_{s=1}^{m} \mathbb{P}^{(\infty)}\left[\max_{s \leq t \leq m} \prod_{\ell=s}^{t-1} \left(\frac{f_\theta(X_\ell|A_\ell)}{f_{\theta_0}(X_\ell|A_\ell)}\right)^{(1-E_\ell)} \geq \beta\right], \qquad (2)$$

by a union bound.

**Lemma 3.** *For each fixed* $s \in [m]$ *and* $\theta \in \Theta^{(1)}$, *the likelihood ratio process* $\left\{\prod_{\ell=s}^{t-1} \left(\frac{f_\theta(X_\ell|A_\ell)}{f_{\theta_0}(X_\ell|A_\ell)}\right)^{(1-E_\ell)}\right\}_{t \geq s}$ *is a mean-1 martingale under the measure* $\mathbb{P}^{(\infty)}$ *and with respect to the filtration* $(\mathcal{F}_t)_{t \geq s}$, *where*

$$\forall t \geq s \quad \mathcal{F}_t := \sigma(E_s, A_s, X_s, E_{s+1}, A_{s+1}, X_{s+1}, \ldots, E_{t-1}, A_{t-1}, X_{t-1}).$$

*Proof.* Taking the conditional expectation of the $t$-th term of the process w.r.t. $\mathcal{F}_{t-1}$, we get

$$\mathbb{E}^{(\infty)}\left[\prod_{\ell=s}^{t-1} \left(\frac{f_\theta(X_\ell|A_\ell)}{f_{\theta_0}(X_\ell|A_\ell)}\right)^{(1-E_\ell)} \bigg| \mathcal{F}_{t-1}\right]$$

$$= \prod_{\ell=s}^{t-2} \left(\frac{f_\theta(X_\ell|A_\ell)}{f_{\theta_0}(X_\ell|A_\ell)}\right)^{(1-E_\ell)} \mathbb{E}^{(\infty)}\left[\left(\frac{f_\theta(X_{t-1}|A_{t-1})}{f_{\theta_0}(X_{t-1}|A_{t-1})}\right)^{(1-E_{t-1})} \bigg| \mathcal{F}_{t-1}\right].$$

The conditional expectation on the right hand side satisfies

$$\mathbb{E}^{(\infty)}\left[\left(\frac{f_\theta(X_{t-1}|A_{t-1})}{f_{\theta_0}(X_{t-1}|A_{t-1})}\right)^{(1-E_{t-1})} \bigg| \mathcal{F}_{t-1}\right]$$

$$= \mathbb{P}^{(\infty)}\left[E_{t-1} = 1\right] + \mathbb{P}^{(\infty)}\left[E_{t-1} = 0\right] \cdot \mathbb{E}^{(\infty)}\left[\frac{f_\theta(X_{t-1}|A_{t-1})}{f_{\theta_0}(X_{t-1}|A_{t-1})} \bigg| \mathcal{F}_{t-1}\right],$$

where we have used the independence of the exploration decision $E_{t-1}$ from the past. By iterated expectation, we have

$$\mathbb{E}^{(\infty)}\left[\frac{f_\theta(X_{t-1}|A_{t-1})}{f_{\theta_0}(X_{t-1}|A_{t-1})} \bigg| \mathcal{F}_{t-1}\right] = \mathbb{E}^{(\infty)}\left[\mathbb{E}^{(\infty)}\left[\frac{f_\theta(X_{t-1}|A_{t-1})}{f_{\theta_0}(X_{t-1}|A_{t-1})} \bigg| A_{t-1}, \mathcal{F}_{t-1}\right] \bigg| \mathcal{F}_{t-1}\right]$$

$$= \mathbb{E}^{(\infty)}\left[1 \mid \mathcal{F}_{t-1}\right],$$

establishing the result. $\qquad\qquad\square$

Thanks to Lemma 3, we can apply Doob's maximal inequality to the non-negative likelihood ratio martingale above to get

$$\mathbb{P}^{(\infty)}\left[\max_{s \leq t \leq m} \prod_{\ell=s}^{t-1} \left(\frac{f_\theta(X_\ell|A_\ell)}{f_{\theta_0}(X_\ell|A_\ell)}\right)^{(1-E_\ell)} \geq \beta\right] \leq \frac{1}{\beta}.$$

Together with (2), this implies

$$\mathbb{P}^{(\infty)}\left[\tau < m\right] \leq \sum_{\theta \in \Theta^{(1)}} \sum_{s=1}^{m} \mathbb{P}^{(\infty)}\left[\max_{s \leq t \leq m} \prod_{\ell=s}^{t-1} \left(\frac{f_\theta(X_\ell|A_\ell)}{f_{\theta_0}(X_\ell|A_\ell)}\right)^{(1-E_\ell)} \geq \beta\right] \leq \frac{|\Theta^{(1)}|\,m}{\beta}.$$

Therefore, using a stopping threshold satisfying $\beta \geq \frac{m|\Theta^{(1)}|}{\alpha}$ guarantees the false alarm property $\mathbb{P}^{(\infty)}\left[\tau < m\right] \leq \alpha$.

## B.2 Detection delay

**Preliminaries.** Let the true post-change parameter starting from an arbitrary time $\nu \in \mathbb{N}$ be equal to $\theta^* \in \Theta^{(1)}$.

Recall that the $\epsilon$-GCD algorithm (Algorithm 1) makes, at round $t \geq 1$, the generalized maximum likelihood estimate

$$\hat{\theta}_t = \operatorname*{argmax}_{\theta \in \Theta^{(1)}} \max_{v=1}^{t} \prod_{\ell=1}^{v-1} f_{\theta_0}(X_\ell | A_\ell)^{E_\ell} \prod_{\ell=v}^{t-1} f_\theta(X_\ell | A_\ell)^{E_\ell}$$

$$= \operatorname*{argmax}_{\theta \in \Theta^{(1)}} \max_{v=1}^{t} \prod_{\ell=1}^{v-1} f_{\theta_0}(X_\ell | A_\ell)^{E_\ell} \prod_{\ell=v}^{t-1} f_\theta(X_\ell | A_\ell)^{E_\ell} \bigg/ \prod_{\ell=1}^{t-1} f_{\theta_0}(X_\ell | A_\ell)^{E_\ell}$$

$$= \operatorname*{argmax}_{\theta \in \Theta^{(1)}} \max_{v=1}^{t} \prod_{\ell=v}^{t-1} \left( \frac{f_\theta(X_\ell | A_\ell)}{f_{\theta_0}(X_\ell | A_\ell)} \right)^{E_\ell} = \operatorname*{argmax}_{\theta \in \Theta^{(1)}} \max_{v=1}^{t} J_{v,t-1}(\theta) = \operatorname*{argmax}_{\theta \in \Theta^{(1)}} \max_{v=1}^{t} \log J_{v,t-1}(\theta),$$

where we have denoted $J_{v,t-1}(\theta) := \prod_{\ell=v}^{t-1} \left( \frac{f_\theta(X_\ell | A_\ell)}{f_{\theta_0}(X_\ell | A_\ell)} \right)^{E_\ell} \Leftrightarrow \log J_{v,t-1}(\theta) = \sum_{\ell=v}^{t-1} E_\ell \log \frac{f_\theta(X_\ell | A_\ell)}{f_{\theta_0}(X_\ell | A_\ell)} = \sum_{\ell=v}^{t-1} U_\ell(\theta)$, where we have defined $U_\ell(\theta) \equiv U_\ell(\theta, \theta_0, X_\ell, A_\ell) := E_\ell \log \frac{f_\theta(X_\ell | A_\ell)}{f_{\theta_0}(X_\ell | A_\ell)}$. Also recall that $Q_t^{(1)}(\theta) = \max_{v=1}^{t} \log J_{v,t-1}(\theta) \geq 0$ represents the CUSUM-style 'exploration queue' statistic for each candidate parameter $\theta \in \Theta^{(1)}$.

**Step 1: Bounding the time for the 'right CUSUM queue $Q_t^{(1)}(\theta^*)$ to outstrip other queues.**

Suppose $\theta \in \Theta^{(1)}$ satisfies $a_\theta^* \neq a_{\theta^*}^*$. For an arbitrary time $t \geq \nu$, we can write

$$\mathbb{P}^{(\nu, \theta^*)} \left[ \max_{i=1}^{t} \log J_{i,t-1}(\theta) \geq \max_{i=1}^{t} \log J_{i,t-1}(\theta^*) \right] = \mathbb{P}^{(\nu, \theta^*)} \left[ \max_{i=1}^{t} \sum_{\ell=i}^{t-1} U_\ell(\theta) \geq \max_{i=1}^{t} \sum_{\ell=i}^{t-1} U_\ell(\theta^*) \right]$$

$$\leq \mathbb{P}^{(\nu, \theta^*)} \left[ \max_{i=1}^{\nu} \sum_{\ell=i}^{\nu-1} U_\ell(\theta) + \max_{i=\nu}^{t} \sum_{\ell=i}^{t-1} U_\ell(\theta) \geq \max_{i=1}^{t} \sum_{\ell=i}^{t-1} U_\ell(\theta^*) \right] \quad \text{(by Lemma 4)}$$

$$\leq \mathbb{P}^{(\nu, \theta^*)} \left[ \max_{i=1}^{\nu} \sum_{\ell=i}^{\nu-1} U_\ell(\theta) + \max_{i=\nu}^{t} \sum_{\ell=i}^{t-1} U_\ell(\theta) \geq \sum_{\ell=\nu}^{t-1} U_\ell(\theta^*) \right] \quad \text{(since } \nu \in [t])$$

$$= \mathbb{P}^{(\nu, \theta^*)} \left[ Q_\nu^{(1)}(\theta) + \max_{i=\nu}^{t} \sum_{\ell=i}^{t-1} U_\ell(\theta) \geq \sum_{\ell=\nu}^{t-1} U_\ell(\theta^*) \right]$$

$$= \mathbb{E}^{(\nu, \theta^*)} \left[ \mathbb{P}^{(\nu, \theta^*)} \left[ Q_\nu^{(1)}(\theta) + \max_{i=\nu}^{t} \sum_{\ell=i}^{t-1} U_\ell(\theta) \geq \sum_{\ell=\nu}^{t-1} U_\ell(\theta^*) \mid Q_\nu^{(1)}(\theta) \right] \right].$$

$$= \mathbb{E}^{(\infty)} \left[ \mathbb{P}^{(\nu, \theta^*)} \left[ Q_\nu^{(1)}(\theta) + \max_{i=\nu}^{t} \sum_{\ell=i}^{t-1} U_\ell(\theta) \geq \sum_{\ell=\nu}^{t-1} U_\ell(\theta^*) \mid Q_\nu^{(1)}(\theta) \right] \right]. \tag{3}$$

The final equality is due to the fact that the inner conditional probability is a function of only $Q_\nu^{(1)}(\theta)$, whose distribution is identical under $\mathbb{P}^{(\nu, \theta^*)}$ and $\mathbb{P}^{(\infty)}$ because it is determined by actions and observations before the change time $\nu$.

We now make the crucial observation that for any $q \geq 0$,

$$\mathbb{P}^{(\nu, \theta^*)} \left[ Q_\nu(\theta) + \max_{i=\nu}^{t} \sum_{\ell=i}^{t-1} U_\ell(\theta) \geq \sum_{\ell=\nu}^{t-1} U_\ell(\theta^*) \mid Q_\nu(\theta) = q \right]$$

$$= \mathbb{P}^{(\nu, \theta^*)} \left[ q + \max_{i=\nu}^{t} \sum_{\ell=i}^{t-1} U_\ell(\theta) \geq \sum_{\ell=\nu}^{t-1} U_\ell(\theta^*) \mid Q_\nu(\theta) = q \right]$$

$$= \mathbb{P}^{(1, \theta^*)} \left[ q + \max_{i=1}^{t-\nu+1} \sum_{\ell=i}^{t-\nu} U_\ell(\theta) \geq \sum_{\ell=1}^{t-\nu} U_\ell(\theta^*) \right]. \tag{4}$$

The first equality above is by simply substituting for $Q_\nu(\theta)$, but the second equality holds because (a) the random variables $E_\ell, X_\ell, A_\ell$ for $\ell \geq \nu$ are independent of $Q_\nu(\theta)$, by virtue of the independent forced exploration enforced in the algorithm, and (b) the probability distribution of exploration actions and their corresponding observations from round $\nu$ onward under $\mathbb{P}^{(\nu,\theta^*)}$ is the same as that of the observations and actions from round 1 onward under $\mathbb{P}^{(1,\theta^*)}$. In other words, *we have rewound the time clock so that the change point is at time* 1 *instead of time* $\nu \geq 1$.

Going forward, to lighten notation, we use $\mathbb{E}$ and $\mathbb{P}$ instead of $\mathbb{E}^{(1,\theta^*)}$ and $\mathbb{P}^{(1,\theta^*)}$ in our calculations. We start by bounding the expectation on the right hand side of (4):

$$\mathbb{P}\left[q + \max_{i=1}^{t-\nu+1} \sum_{\ell=i}^{t-\nu} U_\ell(\theta) \geq \sum_{\ell=1}^{t-\nu} U_\ell(\theta^*)\right]$$
$$\leq \mathbb{P}\left[\psi \geq \sum_{\ell=1}^{t-\nu} U_\ell(\theta^*)\right] + \mathbb{P}\left[q + \max_{i=1}^{t-\nu+1} \sum_{\ell=i}^{t-\nu} U_\ell(\theta) \geq \psi\right], \tag{5}$$

where $\psi$ is chosen as follows. We first introduce the shorthand $D_{\theta_1,\theta_2}(a) := D\left(\theta_1(a)||\theta_2(a)\right)$. We then define, for each $\theta \in \Theta^{(1)} \cup \{\theta_0\}$,

$$\bar{D}_{\theta^*,\theta} := \mathbb{E}^{(1,\theta^*)}\left[D_{\theta^*,\theta}(A_\ell) \mid E_\ell = 1\right] = \sum_{a \in \mathcal{A}} \pi(a) D_{\theta^*,\theta}(a)$$

to be the average KL divergence between $\theta^*$ and $\theta$, when an action is randomly sampled from the exploration distribution $\pi$. Intuitively, $\epsilon(\bar{D}_{\theta^*,\theta_0} - \bar{D}_{\theta^*,\theta})$ is the average rate of drift of the queue $Q_t^{(1)}(\theta)$ at any time after the changepoint $t \geq \nu$ within the exploration rounds; thus, the queue $Q_t^{(1)}(\theta^*)$ enjoys the highest possible drift rate upward.

Finally, we let

$$\psi := \epsilon(t-\nu) \cdot \frac{\bar{D}_{\theta^*,\theta_0} + \left(\bar{D}_{\theta^*,\theta_0} - \bar{D}_{\theta^*,\theta}\right)^+}{2}.$$

We will also find it useful to define the 'gap' of a parameter $\theta$ w.r.t. the true post-change parameter $\theta^*$ as

$$\Delta_\theta := \bar{D}_{\theta^*,\theta_0} - \frac{\bar{D}_{\theta^*,\theta_0} + \left(\bar{D}_{\theta^*,\theta_0} - \bar{D}_{\theta^*,\theta}\right)^+}{2} = \frac{1}{2}\min\left(\bar{D}_{\theta^*,\theta_0}, \bar{D}_{\theta^*,\theta}\right).$$

Note that $0 < \Delta_\theta \leq \frac{\bar{D}_{\theta^*,\theta_0}}{2}$ by Assumption 2.

To bound the first term on the right of (5), we introduce the notation $W_\ell(\theta) \equiv W_\ell(\theta, \theta_0, \theta^*, X_\ell, A_\ell) := \log \frac{f_\theta(X_\ell|A_\ell)}{f_{\theta_0}(X_\ell|A_\ell)} + D_{\theta^*,\theta}(A_\ell) - D_{\theta^*,\theta_0}(A_\ell)$. With this we can write

$$\mathbb{P}\left[\psi \geq \sum_{\ell=1}^{t-\nu} U_\ell(\theta^*)\right] = \mathbb{P}\left[\psi \geq \sum_{\ell=1}^{t-\nu} E_\ell\left(W_\ell(\theta^*) + D_{\theta^*,\theta_0}(A_\ell)\right)\right]$$
$$= \mathbb{P}\left[\sum_{\ell=1}^{t-\nu} E_\ell W_\ell(\theta^*) + E_\ell D_{\theta^*,\theta_0}(A_\ell) < \epsilon(t-\nu)(\bar{D}_{\theta^*,\theta_0} - \Delta_\theta)\right]$$
$$\leq \exp\left(-\frac{\epsilon^2(t-\nu)^2\Delta_\theta^2}{2(r + D_{\max}^2)(t-\nu)}\right) = \exp\left(-\frac{\epsilon^2(t-\nu)\Delta_\theta^2}{2(r + D_{\max}^2)}\right), \tag{6}$$

by a standard Chernoff bound for sums of iid subgaussian random variables; this is because each iid random variable $E_\ell W_\ell(\theta^*) + E_\ell D_{\theta^*,\theta_0}(A_\ell)$ is subgaussian with (variance) parameter $r + D_{\max}^2$ and mean $\epsilon \bar{D}_{\theta^*,\theta_0}$.

On the other hand, the second term on the right side of (5) can be bounded as follows:

$$\mathbb{P}\left[q + \max_{i=1}^{t-\nu+1}\sum_{\ell=i}^{t-\nu} U_\ell(\theta) \geq \psi\right] = \mathbb{P}\left[\max_{i=1}^{t-\nu+1}\sum_{\ell=i}^{t-\nu} E_\ell\left(W_\ell(\theta) + D_{\theta^*,\theta_0}(A_\ell) - D_{\theta^*,\theta}(A_\ell)\right) > \psi - q\right]$$

$$= \mathbb{P}\left[\max_{i=1}^{t-\nu+1}\sum_{\ell=i}^{t-\nu} E_\ell W_\ell(\theta) + \left(E_\ell D_{\theta^*,\theta_0}(A_\ell) - \epsilon \bar{D}_{\theta^*,\theta_0}\right) + \left(-E_\ell D_{\theta^*,\theta}(A_\ell) + \epsilon \bar{D}_{\theta^*,\theta}\right)\right.$$

$$\left. + \epsilon \bar{D}_{\theta^*,\theta_0} - \epsilon \bar{D}_{\theta^*,\theta} > \psi - q\right].$$

The first three terms of each summand above are zero mean and subgaussian with a total variance parameter of $r + 2D_{\max}^2$. Denoting their sum by $D_\ell := E_\ell W_\ell(\theta) + \left(E_\ell D_{\theta^*,\theta_0}(A_\ell) - \epsilon \bar{D}_{\theta^*,\theta_0}\right) + \left(-E_\ell D_{\theta^*,\theta}(A_\ell) + \epsilon \bar{D}_{\theta^*,\theta}\right)$, we split the analysis into two cases.

*Case 1.* If $\bar{D}_{\theta^*,\theta_0} - \bar{D}_{\theta^*,\theta} > 0$, then

$$\mathbb{P}\left[\max_{i=1}^{t-\nu+1}\sum_{\ell=i}^{t-\nu}(D_\ell + \epsilon \bar{D}_{\theta^*,\theta_0} - \epsilon \bar{D}_{\theta^*,\theta}) > \psi - q\right]$$

$$\leq \mathbb{P}\left[\max_{i=1}^{t-\nu+1}\sum_{\ell=i}^{t-\nu} D_\ell + \max_{i=1}^{t-\nu+1}\sum_{\ell=i}^{t-\nu}\epsilon(\bar{D}_{\theta^*,\theta_0} - \bar{D}_{\theta^*,\theta}) > \psi - q\right]$$

$$= \mathbb{P}\left[\max_{i=1}^{t-\nu+1}\sum_{\ell=i}^{t-\nu} D_\ell + \epsilon(t-\nu)(\bar{D}_{\theta^*,\theta_0} - \bar{D}_{\theta^*,\theta}) > \epsilon(t-\nu)\left(\bar{D}_{\theta^*,\theta_0} - \frac{\bar{D}_{\theta^*,\theta}}{2}\right) - q\right]$$

$$= \mathbb{P}\left[\max_{i=1}^{t-\nu+1}\sum_{\ell=i}^{t-\nu} D_\ell > \epsilon(t-\nu)\Delta_\theta - q\right] \leq \exp\left(-\frac{(\epsilon(t-\nu)\Delta_\theta - q)^2}{2(r + 2D_{\max}^2)(t-\nu)}\right), \qquad (7)$$

thanks to Hoeffding's maximal inequality whenever $\epsilon(t-\nu)\Delta_\theta > q$, see e.g., [Jam+14].

*Case 2.* If $\bar{D}_{\theta^*,\theta_0} - \bar{D}_{\theta^*,\theta} \leq 0$, then

$$\mathbb{P}\left[\max_{i=1}^{t-\nu+1}\sum_{\ell=i}^{t-\nu}(D_\ell + \epsilon \bar{D}_{\theta^*,\theta_0} - \epsilon \bar{D}_{\theta^*,\theta}) > \psi - q\right]$$

$$\leq \mathbb{P}\left[\max_{i=1}^{t-\nu+1}\sum_{\ell=i}^{t-\nu} D_\ell + \max_{v=1}^{t-1}\sum_{\ell=v}^{t-1}\epsilon(\bar{D}_{\theta^*,\theta_0} - \bar{D}_{\theta^*,\theta}) > \psi - q\right]$$

$$= \mathbb{P}\left[\max_{i=1}^{t-\nu+1}\sum_{\ell=i}^{t-\nu} D_\ell > \epsilon(t-\nu)\Delta_\theta - q\right] \leq \exp\left(-\frac{(\epsilon(t-\nu)\Delta_\theta - q)^2}{2(r + 2D_{\max}^2)(t-\nu)}\right), \qquad (8)$$

again thanks to Hoeffding's maximal inequality whenever $\epsilon(t-\nu)\Delta_\theta > q$.

Define the following 'bad' event $B_t$, representing the situation that the queue $Q_t^{(1)}(\theta^*)$ has not overtaken some other queue $Q_t^{(1)}(\theta)$ by time $t$:

$$B_t := \bigcup_{\theta:a_\theta^* \neq a_{\theta^*}^*}\left\{\max_{v=1}^{t-1}\log J_{v,t-1}(\theta) \geq \max_{v=1}^{t-1}\log J_{v,t-1}(\theta^*)\right\}.$$

Collecting (3)-(8), and employing an additional union bound over $\Theta^{(1)}$, gives $\forall t \geq \nu$:

$$\mathbb{P}^{(\nu,\theta^*)}\left[B_t \mid Q_\nu^{(1)}\right] \leq$$

$$\sum_{\theta:a_\theta^* \neq a_{\theta^*}^*}\left\{\exp\left(-\frac{\epsilon^2(t-\nu)\Delta_\theta^2}{2(r + D_{\max}^2)}\right) + \mathbf{1}\{\kappa_{t-\nu}(\theta)^c\} + \mathbf{1}\{\kappa_{t-\nu}(\theta)\}\exp\left(-\frac{\left(\epsilon(t-\nu)\Delta_\theta - Q_\nu^{(1)}(\theta)\right)^2}{2(t-\nu)(r + 2D_{\max}^2)}\right)\right\}$$

$$(9)$$

where we have defined $Q_\nu^{(1)} \equiv \left(Q_\nu^{(1)}(\theta)\right)_{\theta \in \Theta^{(1)}}$ to be the set of all CUSUM exploration statistics across parameters, at the beginning of round $\nu$, and denoted $\kappa_{t-\nu}(\theta) := \left\{ Q_\nu^{(1)}(\theta) < \epsilon(t-\nu)\Delta_\theta \right\}$.

Note that by the definition of the algorithm, we have $B_t^c \cap \{E_t = 0\} \subseteq \{A_t = a_{\theta^*}^*\}$.

**Step 2: Bounding the additional time for the CUSUM queue $Q_t^{(2)}(\theta^*)$ to rise above the threshold and trigger stopping.**

For each $\theta \in \Theta^{(1)}$ and $s, t \in \mathbb{N}$, recall the exploitation-based CUSUM statistic for $\theta_0$ versus $\theta$ [Lor71], after having accumulated $t$ rounds worth of observations in exploitation phases:

$$Q_{t+1}^{(2)}(\theta) := \log \max_{1 \le s \le t+1} \prod_{\ell=s}^t \left( \frac{f_\theta(X_\ell|A_\ell)}{f_{\theta_0}(X_\ell|A_\ell)} \right)^{1-E_\ell} = \max_{1 \le s \le t+1} \sum_{\ell=s}^t (1-E_\ell) \log \left( \frac{f_\theta(X_\ell|A_\ell)}{f_{\theta_0}(X_\ell|A_\ell)} \right),$$

where the empty product is defined to be $1$, as usual. This statistic satisfies the recursive relation

$$Q_{t+1}^{(2)}(\theta) = \left( Q_t^{(2)}(\theta) + (1-E_t) \log \left( \frac{f_\theta(X_t|A_t)}{f_{\theta_0}(X_t|A_t)} \right) \right)^+.$$

Moreover, the algorithm stops as soon as $\max_{\theta \in \Theta^{(1)}} Q_t^{(2)}(\theta)$ exceeds the level $\log \beta$.

To lighten our notational burden, we henceforth use $\mathbb{P}_Q^{(\nu,\theta^*)}$ to denote the conditional measure $\mathbb{P}^{(\nu,\theta^*)}[\cdot \mid Q_\nu^{(1)}]$. We have, for any positive integer $k$, that

$$\mathbb{P}_Q^{(\nu,\theta^*)}[(\nu,\theta^*)] \tau \ge \nu + k \le \mathbb{P}_Q^{(\nu,\theta^*)} \left[ \tau \ge \nu + k, \bigcap_{t=\nu+\frac{k}{2}}^{\nu+k} B_t^c \right] + \mathbb{P}_Q^{(\nu,\theta^*)} \left[ \bigcup_{t=\nu+\frac{k}{2}}^{\nu+k} B_t \right]. \qquad (10)$$

Also, by the definition of $\tau$ and the maximum-of-partial-sums property of $Q_t^{(2)}(\theta)$, we have

$$\mathbb{P}_Q^{(\nu,\theta^*)} \left[ \tau \ge \nu + k, \bigcap_{t=\nu+\frac{k}{2}}^{\nu+k} B_t^c \right] \le \mathbb{P}_Q^{(\nu,\theta^*)} \left[ \sum_{\ell=\nu+\frac{k}{2}}^{\nu+k} (1-E_\ell) \log \left( \frac{f_{\theta^*}(X_\ell|A_\ell)}{f_{\theta_0}(X_\ell|A_\ell)} \right) < \log \beta, \bigcap_{t=\nu+\frac{k}{2}}^{\nu+k} B_t^c \right]$$

$$\le \mathbb{P}_Q^{(\nu,\theta^*)} \left[ \sum_{\ell=\nu+\frac{k}{2}}^{\nu+k} (1-E_\ell) \log \left( \frac{f_{\theta^*}(X_\ell|A_\ell)}{f_{\theta_0}(X_\ell|A_\ell)} \right) < \log \beta, \bigcap_{t=\nu+\frac{k}{2}}^{\nu+k} B_t^c, G_{\nu+k/2,\nu+k} \right] + \mathbb{P}_Q^{(\nu,\theta^*)} \left[ G_{\nu+k/2,\nu+k}^c \right], \qquad (11)$$

where we have defined the 'good' events

$$G_{i,j} := \left\{ \sum_{s=i}^j (1-E_s) \ge \frac{1}{2}(j-i+1)(1-\epsilon) \right\}$$

for any $i, j$, representing an adequate amount of exploitation in the time interval $\{i, i+1, \ldots, j\}$. Assuming $\epsilon \le 1/2$, by Hoeffding's inequality we get

$$\mathbb{P}_Q^{(\nu,\theta^*)} \left[ G_{\nu+k/2,\nu+k}^c \right] \le e^{-k/16}. \qquad (12)$$

By the law of total probability, we have

$$\mathbb{P}_Q^{(\nu,\theta^*)} \left[ \sum_{\ell=\nu+\frac{k}{2}}^{\nu+k} (1-E_\ell) \log \left( \frac{f_{\theta^*}(X_\ell|A_\ell)}{f_{\theta_0}(X_\ell|A_\ell)} \right) < \log \beta, \bigcap_{t=\nu+\frac{k}{2}}^{\nu+k} B_t^c, G_{\nu+k/2,\nu+k} \right]$$

$$= \sum_{j=\frac{1}{4}k(1-\epsilon)}^{k/2} \mathbb{P}_Q^{(\nu,\theta^*)} \left[ \sum_{s=\nu+k/2}^{\nu+k} (1-E_s) = j, \bigcap_{t=\nu+k/2}^{\nu+k} B_t^c \right] \times$$

$$\mathbb{P}_Q^{(\nu,\theta^*)} \left[ \sum_{\ell=\nu+k/2}^{\nu+k} (1-E_\ell) \log \left( \frac{f_{\theta^*}(X_\ell|A_\ell)}{f_{\theta_0}(X_\ell|A_\ell)} \right) < \log \beta \,\middle|\, \sum_{s=\nu+k/2}^{\nu+k} (1-E_s) = j, \bigcap_{t=\nu+k/2}^{\nu+k} B_t^c \right]. \qquad (13)$$

Recall that under $\mathbb{P}_Q^{(\nu,\theta^*)}[\cdot]$, for any $\ell \geq \nu$, the random variable $\log \frac{f_{\theta^*}(X_\ell|A_\ell)}{f_{\theta_0}(X_\ell|A_\ell)}$ is $r$-subgaussian and has mean $D\left(\theta^*(A_\ell)||\theta_0(A_\ell)\right)$, conditioned on the past trajectory up to and including $A_\ell$.

Consider now an alternative (and equivalent) probability space where the sequence of observations from playing the action $a_{\theta^*}^*$ in any exploitation round not earlier than $\nu + k/2$ (i.e., any round index $\ell \geq \nu + k/2$ with $E_\ell = 0$) is revealed sequentially in order from the iid sequence $Z_1, Z_2, \ldots$, where each $Z_i$ has the probability distribution $\mathbb{P}^{(1,\theta^*)}\left[\log \frac{f_{\theta^*}(X|a_{\theta^*}^*)}{f_{\theta_0}(X|a_{\theta^*}^*)}\right]$. Define $\mu^* := D\left(\theta^*(a_{\theta^*}^*)||\theta_0(a_{\theta^*}^*)\right)$ to be the mean of each $Z_i$.

We invoke standard subgaussian concentration of iid sums in this equivalent probability space, say $\tilde{\mathbb{P}}$, to get

$$\mathbb{P}_Q^{(\nu,\theta^*)}\left[\sum_{\ell=\nu+k/2}^{\nu+k}(1-E_\ell)\log\left(\frac{f_{\theta^*}(X_\ell|A_\ell)}{f_{\theta_0}(X_\ell|A_\ell)}\right) < \log\beta \,\middle|\, \sum_{s=\nu+k/2}^{\nu+k}(1-E_s)=j, \bigcap_{t=\nu+k/2}^{\nu+k}B_t^c\right]$$

$$= \tilde{\mathbb{P}}\left[\sum_{i=1}^{j}Z_i < \log\beta \,\middle|\, \sum_{s=\nu+k/2}^{\nu+k}(1-E_s)=j, \bigcap_{t=\nu+k/2}^{\nu+k}B_t^c\right]$$

$$= \tilde{\mathbb{P}}\left[\sum_{i=1}^{j}Z_i < \log\beta\right] \quad (\because \{Z_i\}_i \text{ depend only on exploitation outcomes}, \{B_t\}_t \text{ depend only on exploration outcomes})$$

$$\leq \exp\left(-\frac{(\log\beta - j\mu^*)^2}{2jr}\right)$$

whenever $j \geq \frac{\log\beta}{\mu^*}$, by a Chernoff bound. Using this in (13) gives

$$\mathbb{P}_Q^{(\nu,\theta^*)}\left[\sum_{\ell=\nu+k/2}^{\nu+k}(1-E_\ell)\log\left(\frac{f_{\theta^*}(X_\ell|A_\ell)}{f_{\theta_0}(X_\ell|A_\ell)}\right) < \log\beta, \bigcap_{t=\nu+k/2}^{\nu+k}B_t^c, G_{\nu+k/2,\nu+k}\right]$$

$$\leq \max_{j=\frac{1}{4}k(1-\epsilon)}^{k/2}\exp\left(-\frac{(\log\beta-j\mu^*)^2}{2jr}\right) = \exp\left(-\frac{(\log\beta-j^*\mu^*)^2}{2j^*r}\right) \tag{14}$$

with $j^* := \frac{k}{4}(1-\epsilon)$, as long as $j^* \geq \frac{\log\beta}{\mu^*} \Leftrightarrow k \geq \frac{4\log\beta}{\mu^*(1-\epsilon)}$. This follows by the fact that the function $x \mapsto \frac{x^2}{a+x}$ with $a > 0$ is increasing in $(0,\infty)$.

**Step 3. Putting together the time bounds to get an overall delay bound.**

Putting together (9)-(14) and denoting $\gamma := r + D_{\max}^2$, we get that whenever $k \geq \frac{4\log\beta}{\mu^*(1-\epsilon)}$,

$$\mathbb{P}_Q^{(\nu,\theta^*)}[\tau \geq \nu + k]$$

$$\leq \sum_{\theta:a_\theta^* \neq a_{\theta^*}^*}\sum_{t=\nu+k/2}^{\nu+k}\left\{\exp\left(-\frac{\epsilon^2(t-\nu)\Delta_\theta^2}{2\gamma}\right) + \mathbf{1}\left\{\kappa_{t-\nu}(\theta)^c\right\} + \mathbf{1}\left\{\kappa_{t-\nu}(\theta)\right\}e^{-\frac{\left(\epsilon(t-\nu)\Delta_\theta - Q_\nu^{(1)}(\theta)\right)^2}{2(t-\nu)\gamma}}\right\}$$

$$+ e^{-k/16} + e^{-\frac{(\log\beta - j^*\mu^*)^2}{2j^*r}}$$

$$\leq \sum_{\theta:a_\theta^* \neq a_{\theta^*}^*}\left\{\frac{e^{-\frac{k\epsilon^2\Delta_\theta^2}{4\gamma}}}{\left(1-e^{-\frac{\epsilon^2\Delta_\theta^2}{2\gamma}}\right)} + 1 \wedge \sum_{t=\nu+k/2}^{\nu+k}\left(\mathbf{1}\left\{\kappa_{t-\nu}(\theta)^c\right\} + \mathbf{1}\left\{\kappa_{t-\nu}(\theta)\right\}e^{-\frac{\left(\epsilon(t-\nu)\Delta_\theta - Q_\nu^{(1)}(\theta)\right)^2}{2(t-\nu)\gamma}}\right)\right\}$$

$$+ e^{-k/16} + e^{-\frac{\left(\log\beta - \frac{k(1-\epsilon)\mu^*}{4}\right)^2}{\frac{k}{2}(1-\epsilon)r}}, \tag{15}$$

where we have denoted $1 \wedge x := \min\{x,1\}$, and we have taken the minimum of the inner sum (over $t$) with 1 because probabilities are always bounded by 1.

The inner sum above for a fixed $\theta$, clamped to 1, can be bounded as follows. Let $s_0 := \frac{2Q_\nu^{(1)}(\theta)}{\epsilon\Delta_\theta}$, so that $\epsilon s\Delta_\theta - Q_\nu^{(1)}(\theta) \geq \frac{\epsilon s\Delta_\theta}{2}$ whenever $s \geq s_0$. So $\forall\theta$, denoting $\gamma := r+2D_{\max}^2$, and $1 \wedge x := \min\{x,1\}$

we have

$$1 \wedge \sum_{t=\nu+k/2}^{\nu+k} \left( \mathbf{1}\left\{\kappa_{t-\nu}(\theta)^c\right\} + \mathbf{1}\left\{\kappa_{t-\nu}(\theta)\right\} e^{-\frac{\left(\epsilon(t-\nu)\Delta_\theta - Q_\nu^{(1)}(\theta)\right)^2}{2(t-\nu)\gamma}} \right)$$

$$= 1 \wedge \sum_{s=k/2}^{k} \left( \mathbf{1}\left\{\kappa_s(\theta)^c\right\} + \mathbf{1}\left\{\kappa_s(\theta)\right\} e^{-\frac{\left(\epsilon s\Delta_\theta - Q_\nu^{(1)}(\theta)\right)^2}{2s\gamma}} \right)$$

$$\leq \mathbf{1}\left\{k/2 \geq s_0\right\} \sum_{s=k/2}^{k} e^{-\frac{\epsilon^2 s\Delta_\theta^2}{8\gamma}} + \mathbf{1}\left\{k/2 < s_0\right\} \cdot 1$$

$$\leq \mathbf{1}\left\{k \geq \frac{4Q_\nu^{(1)}(\theta)}{\epsilon\Delta_\theta}\right\} \left( \frac{e^{-\frac{k\epsilon^2\Delta_\theta^2}{16\gamma}}}{1 - e^{-\frac{\epsilon^2\Delta_\theta^2}{8\gamma}}} \right) + \mathbf{1}\left\{k < \frac{4Q_\nu^{(1)}(\theta)}{\epsilon\Delta_\theta}\right\} \cdot 1. \tag{16}$$

We are now in a position to obtain a bound on the (conditional) expected excess detection delay $\mathbb{E}^{(\nu,\theta^*)}\left[(\tau-\nu)^+\right]$ by integrating the tail and using (15) and (16):

$$\mathbb{E}_Q^{(\nu,\theta^*)}\left[(\tau-\nu)^+\right] = \sum_{k=1}^{\infty} \mathbb{P}_Q^{(\nu,\theta^*)}\left[(\tau-\nu)^+ \geq k\right] = \sum_{k=1}^{\infty} \mathbb{P}_Q^{(\nu,\theta^*)}\left[\tau \geq \nu+k\right]$$

$$\leq 20 + \frac{4\log\beta}{\mu^*(1-\epsilon)} + \sum_{k=\left\lceil\frac{4\log\beta}{\mu^*(1-\epsilon)}\right\rceil}^{\infty} e^{-\frac{\left(\log\beta - \frac{k(1-\epsilon)\mu^*}{4}\right)^2}{\frac{k}{2}(1-\epsilon)r}}$$

$$+ \sum_{\theta:a_\theta^*\neq a_{\theta^*}^*} \left\{ \sum_{k=\left\lceil\frac{4\log\beta}{\mu^*(1-\epsilon)}\right\rceil}^{\infty} \frac{e^{-\frac{k\epsilon^2\Delta_\theta^2}{4\gamma}}}{\left(1 - e^{-\frac{\epsilon^2\Delta_\theta^2}{2\gamma}}\right)} + \frac{4Q_\nu^{(1)}(\theta)}{\epsilon\Delta_\theta} + \sum_{k=\left\lceil\frac{4\log\beta}{\mu^*(1-\epsilon)}\right\rceil}^{\infty} \frac{e^{-\frac{k\epsilon^2\Delta_\theta^2}{16\gamma}}}{\left(1 - e^{-\frac{\epsilon^2\Delta_\theta^2}{8\gamma}}\right)} \right\}.$$

The third term above admits the bound

$$\sum_{k=\left\lceil\frac{4\log\beta}{\mu^*(1-\epsilon)}\right\rceil}^{\infty} e^{-\frac{\left(\log\beta - \frac{k(1-\epsilon)\mu^*}{4}\right)^2}{\frac{k}{2}(1-\epsilon)r}} = \sum_{k=\left\lceil\frac{4\log\beta}{\mu^*(1-\epsilon)}\right\rceil}^{\left\lceil\frac{8\log\beta}{\mu^*(1-\epsilon)}\right\rceil} e^{-\frac{\left(\log\beta - \frac{k(1-\epsilon)\mu^*}{4}\right)^2}{\frac{k}{2}(1-\epsilon)r}} + \sum_{k=\left\lceil\frac{8\log\beta}{\mu^*(1-\epsilon)}\right\rceil}^{\infty} e^{-\frac{\left(\log\beta - \frac{k(1-\epsilon)\mu^*}{4}\right)^2}{\frac{k}{2}(1-\epsilon)r}}$$

$$\leq 1 + \frac{4\log\beta}{\mu^*(1-\epsilon)} + \frac{e^{-\frac{\mu^*\log\beta}{4r}}}{1 - e^{-\frac{(\mu^*)^2(1-\epsilon)}{32r}}},$$

while each summand corresponding to $\theta$ in the final term is bounded as

$$\sum_{k=\left\lceil\frac{4\log\beta}{\mu^*(1-\epsilon)}\right\rceil}^{\infty} \frac{e^{-\frac{k\epsilon^2\Delta_\theta^2}{4\gamma}}}{\left(1 - e^{-\frac{\epsilon^2\Delta_\theta^2}{2\gamma}}\right)} + \frac{4Q_\nu^{(1)}(\theta)}{\epsilon\Delta_\theta} + \sum_{k=\left\lceil\frac{4\log\beta}{\mu^*(1-\epsilon)}\right\rceil}^{\infty} \frac{e^{-\frac{k\epsilon^2\Delta_\theta^2}{16\gamma}}}{\left(1 - e^{-\frac{\epsilon^2\Delta_\theta^2}{8\gamma}}\right)} \leq \frac{4Q_\nu^{(1)}(\theta)}{\epsilon\Delta_\theta} + \frac{2e^{-\frac{\epsilon^2\Delta_\theta^2\log\beta}{4\gamma\mu^*(1-\epsilon)}}}{1 - e^{-\frac{\epsilon^2\Delta_\theta^2}{8\gamma}}},$$

giving

$$\mathbb{E}_Q^{(\nu,\theta^*)}\left[(\tau-\nu)^+\right] \leq 21 + \frac{8\log\beta}{\mu^*(1-\epsilon)} + \sum_{\theta:a_\theta^*\neq a_{\theta^*}^*} \left( \frac{4Q_\nu^{(1)}(\theta)}{\epsilon\Delta_\theta} + \frac{2e^{-\frac{\epsilon^2\Delta_\theta^2\log\beta}{4\gamma\mu^*(1-\epsilon)}}}{1 - e^{-\frac{\epsilon^2\Delta_\theta^2}{8\gamma}}} \right),$$

where (recall) $\mu^* := D\left(\theta^*(a_{\theta^*}^*)||\theta_0(a_{\theta^*}^*)\right)$ and $\gamma = r + 2D_{\max}^2$.

Taking expectation under the $\mathbb{P}^{(\infty)}$ distribution of $Q_\nu$ completes the proof of Theorem 3.

**Lemma 4.** *For any sequence $x_1, \ldots, x_{t-1}$ and $\nu \in [t]$,*

$$\max_{i=1}^{t} \sum_{\ell=i}^{t-1} x_\ell \leq \left( \max_{i=1}^{\nu-1} \sum_{\ell=i}^{\nu-1} x_\ell \right)^+ + \max_{i=\nu}^{t} \sum_{\ell=i}^{t-1} x_\ell = \max_{i=1}^{\nu} \sum_{\ell=i}^{\nu-1} x_\ell + \max_{i=\nu}^{t} \sum_{\ell=i}^{t-1} x_\ell.$$

*Proof of Lemma 4.* The lemma is a consequence of the elementary result that $\max_i(a_i + b_i) \leq \max_i a_i + \max_i b_i$ for any discrete collection of numbers $\{a_i\}_i, \{b_i\}_i$. $\qquad\square$

## C Experiment Details

This section describes in detail the setup and methodology followed for obtaining the results in Section 6. It also includes additional results, both for the synthetic and audio sensing settings, that explore the impact of various problem parameters on performance.

### C.1 Version of the $\epsilon$-GCD algorithm used in experiments

In all our experiments, we used the *full data-MLE* implementation of the $\epsilon$-GCD template, as given in Algorithm 1. The only difference of this algorithm from the *exploration data-MLE* version given in the main text (Algorithm 1) is that the estimate $\hat{\theta}_t$ for the post-change distribution is computed using data from *all* previous rounds, regardless of exploration or exploitation.

---

**Algorithm 1** $\epsilon$-GCD (Full data-MLE version)

---

1: **Input:** Exploration rate $\epsilon \in [0,1]$, Pre-change parameter $\theta_0$, Post-change parameter set $\Theta^{(1)}$, Stopping threshold $\beta > 0$, Exploration distribution over actions $\pi$, Observation function $g_\theta(x, a)$ $\forall (\theta, x, a) \in \Theta^{(1)} \times \mathcal{X} \times \mathcal{A}$.

2: **Init:** $Q_1^{(1)}(\theta) \leftarrow 0, Q_1^{(2)}(\theta) \leftarrow 0 \ \forall \theta \in \Theta^{(1)}$      {CUSUM statistics based on overall and exploitation-only data, per parameter}

3: **for** round $t = 1, 2, 3, \ldots$ **do**

4:     **if** $\max_{\theta \in \Theta^{(1)}} Q_t^{(2)}(\theta) \geq \beta$ **then**

5:        **break**                                      {Stop sampling and exit}

6:     **end if**

7:     Sample $E_t \sim \text{Ber}(\epsilon)$ independently

8:     **if** $E_t == 1$ **then**

9:        Play action $A_t \sim \pi$ independently                           {Explore}

10:       Get observation $X_t$

11:    **else**

12:        Compute $\hat{\theta}_t = \text{argmax}_{\theta \in \Theta^{(1)}} Q_t^{(1)}(\theta)$    {Most likely post-change distribution based on all past data}

13:        Play action $A_t = \text{argmax}_{a \in \mathcal{A}} D\left(\theta^{(0)}(a) || \hat{\theta}_t(a)\right)$

14:       Get observation $X_t$

15:        Set $\forall \theta \in \Theta^{(1)} : Q_{t+1}^{(2)}(\theta) \leftarrow \left(Q_t^{(2)}(\theta) + g_\theta(X_t, A_t)\right)^+$     {Update exploitation CUSUM statistics}

16:    **end if**

17:    Set $\forall \theta \in \Theta^{(1)} : Q_{t+1}^{(1)}(\theta) \leftarrow \left(Q_t^{(1)}(\theta) + g_\theta(X_t, A_t)\right)^+$     {Update overall data CUSUM statistics}

18: **end for**

---

The chief reason to prefer the exploration-only version in the main text is that the theoretical analysis of its detection delay is slightly simpler than the full-data version. This is because the exploration-only CUSUM statistic (queue) changes from one time to the next in an essentially memoryless manner since the sensing action is chosen independent of the past.

On the other hand, we preferred the full data-version in experiments since it was noticed to offer slightly better numerical performance. We remark that the full-data MLE version can also be analyzed rigorously for its detection delay[9] in a manner similar to that of Algorithm 1, with an essentially similar delay guarantee. However, one will have to content with an extra overhead in the additive term of the detection delay (i.e., the last term of (1)), due to not being able to apply time-uniform maximal inequalities for martingales (see (7), (8) in Section B.2) but resort to a slightly worse union bound over time.

### C.2 Algorithms compared in the experiments

We compare the performance of the following adaptive sensing change detection algorithms in all our experimental settings:

---

[9]The false alarm rate analysis remains the same as there is no change to the stopping criterion in the algorithm.

1. The $\epsilon$-GCD algorithm ('EG' in plots), used with $\epsilon = 0.2$

2. Uniform Random Sampling ('URS'): The non-adaptive sensing rule that plays an action drawn uniformly at random from a given set of actions; this is $\epsilon$-GCD with $\epsilon = 0$

3. Oracle Sampling ('Oracle'): The sensing rule that always plays the most informative action knowing the post-change distribution in advance: $\mathrm{argmax}_{a \in \mathcal{A}} D\left(\theta^*(a)||\theta^{(0)}(a)\right)$.

## C.3 Synthetic experiments

We conduct change detection experiments on a line graph, of size $N$, serving as an ambient space. Nodes of the graph are interpreted as physical locations, and take values in $[N]$. Nodes $j, k$ are connected if $|j - k| = 1$. Each node $n \in [N]$ offers a Gaussian-distributed observation depending on the changepoint $\nu \in \mathbb{N}$. In particular, the signal (observations from all nodes) at time $t$ is a random vector $S(t) = (S_n(t))_{n \in [N]} \in \mathbb{R}^N$, where

$$S_n(t) = \theta_n \mathbf{1}\{t \geq \nu\} + W_n(t), \ t = 0, 1, 2, \ldots \tag{17}$$

$\theta := (\theta_n)_{n \in [N]}$ represents the post-change parameter and $W_n(t) \in \mathcal{N}(0, \sigma^2)$ are IID Gaussian random variables for $n \in [N]$ and across time $t = 0, 1, \ldots,,$ and represents observation noise. We also choose $\sigma^2 = 1/2$ for all our synthetic experiments. Note that in essence, this setup has the pre-change parameter set to zero ($\theta = 0 \in \mathbb{R}^N$).

*Isolated and Structured Anomalies.* We consider two types of the vector of change parameters:
(a) Isolated singleton change, namely, $\theta_n \in \{0, 1\}$ and $\sum_{n \in [N]} \theta_n = 1$;
(b) Structured K-change: We consider parameter changes with[10] $|\mathrm{Supp}(\theta)| = K$, and the nodes (components) corresponding to the non-zero support are connected. As such the collection of anomalies is $N - K + 1$ corresponding to different starting positions.

*Diffuse and 'Pointy' Action sets.* In a parallel fashion we allow actions to be vertices of the $N$-hypercube, $a_n \in \mathcal{A} \subset \{0, 1\}^N$, and the action sets to be either *pointy* ($\mathcal{A}_1$), namely, $|\mathrm{Supp}(a)| = 1$ $\forall a \in \mathcal{A}_1$, which allows probing only single nodes, or *diffuse* ($\mathcal{A}_2$), where only a connected subset of nodes can be queried. In either case, the observation received on an action, $a \in \mathcal{A}$ is given by $X_a = \langle \frac{a}{\|a\|_2}, S \rangle$, where we impose the normalization because we want to maintain the same signal-to-noise ratio (SNR) across different types of probes.

We reported results for pointy actions and isolated anomalies in Sec. 6. We will describe experiments with other scenarios here.

**Structured Anomalies and Diffuse Action Sets.**
We experiment with Structured K-change as described above with $K = 5$. The action sets are diffuse:

$$\mathcal{A} = \{a \in \{0, 1\}^N : \|a\|_2 = 1, |\mathrm{Supp}(a)| = 5, \ a \text{ is connected.}\}$$

This means that corresponding to each anomalous change, there is an action (unknown to the learner) that perfectly overlaps with the entire anomalous change. Furthermore, there are several other actions that partially overlap with the structured anomaly. As a result there is a higher probability of detecting anomalies.

As in Sec. 6 we tabulate results for change point $\nu = 40$, for different sizes of graph. Our results are based on 5000 Monte Carlo runs. The mean and standard deviation for change point $\nu = 40$ is reported in Table 2. We see that, although diffuse, such actions appear to improve detection delay for $\epsilon$-GCD in comparison to the case considered in Sec. 6. This is to be expected because the number of structured anomalies are smaller. For instance for a graph of size 10, we only have 5 anomalous parameter changes. Furthermore, we can detect anomalies even when the actions only partially overlap with the anomaly. Thus change detection methods now have a larger probability to detect parameter changes in contrast to isolated anomalies.

We also observe that $\epsilon$-GCD is still as effective, and closely mirrors Oracle performance. We do not tabulate the effect of different changepoints here. This is because, we notice that the changepoint parameter $\nu$ has no noticeable effect when graph size is held constant for all of the reported methods (Oracle, URS, and $\epsilon$-GCD ).

---

[10]We define $\mathrm{Supp}(x) := \{i : x_i \neq 0\}$.

| Size | Oracle | $\epsilon$-GCD(Full) | $\epsilon$-GCD(Theory) | URS |
|---|---|---|---|---|
| 10 | $51 \pm 7$ | $64 \pm 12$ | $90 \pm 21$ | $96 \pm 14$ |
| 15 | $51 \pm 7$ | $71 \pm 13$ | $100 \pm 27$ | $163 \pm 26$ |
| 20 | $51 \pm 7$ | $74 \pm 14$ | $113 \pm 35$ | $237 \pm 40$ |
| 25 | $51 \pm 7$ | $80 \pm 17$ | $126 \pm 46$ | $310 \pm 52$ |

Table 2: Structured Anomalies with Diffuse Actions: Observed mean and standard deviation for the simulated stopped time for varying graph-sizes with diffused action set for change occurring at $\nu = 40$.

**Structured Anomalies with Pointy Action Sets.**
Here we experiment with $K = 5$ as in the setup above but examine the effect of pointy action sets. As a result our actions can only probe some component of the anomaly. Observe that the anomaly is spread across a larger region. Therefore, a pointy anomaly can only sample a small part of the parameter change in any round.

We report mean and variance for expected delay for change point $\nu = 40$, for different graph sizes over 5000 Monte Carlo runs in Table 3.

| Size | Oracle | $\epsilon$-GCD(Full) | $\epsilon$-GCD(Theory) | URS |
|---|---|---|---|---|
| 10 | $150 \pm 27$ | $192 \pm 41$ | $289 \pm 124$ | $299 \pm 57$ |
| 15 | $149 \pm 27$ | $210 \pm 49$ | $340 \pm 175$ | $448 \pm 85$ |
| 20 | $150 \pm 27$ | $204 \pm 44$ | $350 \pm 213$ | $597 \pm 118$ |
| 25 | $149 \pm 27$ | $213 \pm 48$ | $395 \pm 246$ | $746 \pm 145$ |

Table 3: Structured Anomalies with Pointy Action Sets. Observed mean and standard deviation for the simulated stopped time for varying graph-sizes at $\nu = 40$.

In this experiment both Oracle and $\epsilon$-GCD exhibit larger delays. The reason now is that pointy anomalies can only sample a single component, and as such a component in the anomalous region exhibits smaller change, and so it takes a longer time to detect. Again, no noticeable impact of varying changepoint on delay was observed.

**Isolated Anomalies with Diffuse Action Sets.**
Here we consider the case where the anomalies are isolated but the action sets are diffuse. Our results (mean and variance) over 5000 Monte Carlo runs for changepoint $\nu = 40$ is tabulated in Table 4.

| Size | Oracle | $\epsilon$-GCD+ | $\epsilon$-GCD(Theory) | URS |
|---|---|---|---|---|
| 10 | $249 \pm 35$ | $277 \pm 40$ | $321 \pm 49$ | $497 \pm 74$ |
| 15 | $250 \pm 35$ | $281 \pm 40$ | $326 \pm 54$ | $549 \pm 81$ |
| 20 | $249 \pm 35$ | $297 \pm 42$ | $370 \pm 84$ | $998 \pm 151$ |
| 25 | $249 \pm 35$ | $324 \pm 52$ | $509 \pm 237$ | $1044 \pm 157$ |

Table 4: Isolated Anomalies with Diffuse Action Sets. Observed mean and standard deviation for the simulated stopped time for varying graph-sizes for change occurring at $\nu = 40$.

Among all of the different scenarios, this setup has uniformly larger expected delay across all of the methods (Oracle, $\epsilon$-GCD and URS). This is not surprising considering the fact that isolated anomalies when probed with diffuse actions manifest as substantially smaller change. This is because a diffuse action, spread across 5 locations, is capable of collecting only a 5th of the energy of the anomaly.

## D   Experiment Details for Real-World Audio Dataset

Recall from Sec. 6 we explored changepoint detection on the MMII dataset. We pointed out that we used reconstruction errors of auto-encoders, and modeled these errors with Gaussians. Here we provide more details and additional experiments on the dataset.

*Audio Processing.* For each audio-stream we train auto-encoders on normal data using mel-spectrogram features. We train different autoencoders for different machine ids. We use mean of reconstruction errors from each of these machines when there are no anomalies, and construct the pre-change parameter vector. Similarly, we use mean of reconstruction errors from each of these machines when there are anomalies and construct post-change parameters.

To compute reconstruction errors, we adopt the autoencoder architecture as used in Section 4 in [Pur+19b]. We also make use of publicly available code to train autoencoders with mel-spectrogram features of normal data as inputs. We use the same parameters that are used in [Pur+19b], to extract mel-spectrogram features from a given audio input. We assume that, for a given audio stream, there is only one anomaly, and that anomaly is present in only one of the machine ids.

The resulting pre-change parameters across the 4 machines and the post-change parameters under an anomaly are displayed below:

| Machine ID | Mean reconstruction error | |
|:---:|:---:|:---:|
| | Normal | Abnormal |
| 00 | 7.816003 | 18.043417 |
| 02 | 7.728631 | 12.879204 |
| 04 | 12.029381 | 15.425252 |
| 06 | 9.34813 | 10.788003 |

Table 5: Mean reconstruction errors for machine ID $00, 02, 04, 06$ under normal and abnormal operation

Using the notation of our synthetic experiment, our setup here can be described as the case with isolated anomalies (i.e., only one machine has an anomaly), and pointy actions (i.e., we can only query one machine at any time).

*Experiment.* In addition to the $\nu = 6$ case, which we reported in the main paper, we simulate changepoints for $\nu = 21$. This corresponds to 210 seconds. We do this by introducing anomalies in machine bearing ID00 as follows. We concatenated 21 normal files and 39 abnormal files chosen uniformly at random from machine ID00. For the other machines we concatenated 60 normal files at random. The 60 files correspond to 600 seconds. Our changepoint corresponds to 21st file, which we denote as $\nu = 21$ and our task is to detect this change. Note that both the machine ID and the changepoint is not known to the learner. Our results are depicted as histograms for changepoints of anomaly detection in Fig. 4.

As observed Oracle method has a small variance, and the histogram is concentrated at around 25, which is about 40 seconds delay. $\epsilon$-GCD also exhibits small delay, but its variance is somewhat larger in this context. URS expected delay and variance are substantially larger. This demostrates the gains due to adaptive processing.

## E   Detailed Survey of Related Work

Changepoint detection deals with identifying points in time when probability laws governing a stochastic process changes abruptly. The problem of changepoint detection has been widely studied, dating back to the pioneering works of Page [Pag54] and Lorden [Lor71].

Online change detection focuses on situations where the data is obtained incrementally over time, and one must infer whether a change has occurred at each time. A large part of the online change detection literature, like our paper, adopts a frequentist approach, and, in particular, utilizes parametric models for pre-change and post-change distributions. In this context, the CUSUM algorithm [Pag54] and its variants such as the generalized likelihood ratio statistics have been proposed, in which a change is announced when the likelihood ratio statistic exceeds a threshold. While there are a number of prior works on this topic (see [BN93; CG12; VB14]), specific attention to finite time (i.e., non-asymptotic) guarantees on detection delay is more recent [LX10; Mai19], and as such remains somewhat open

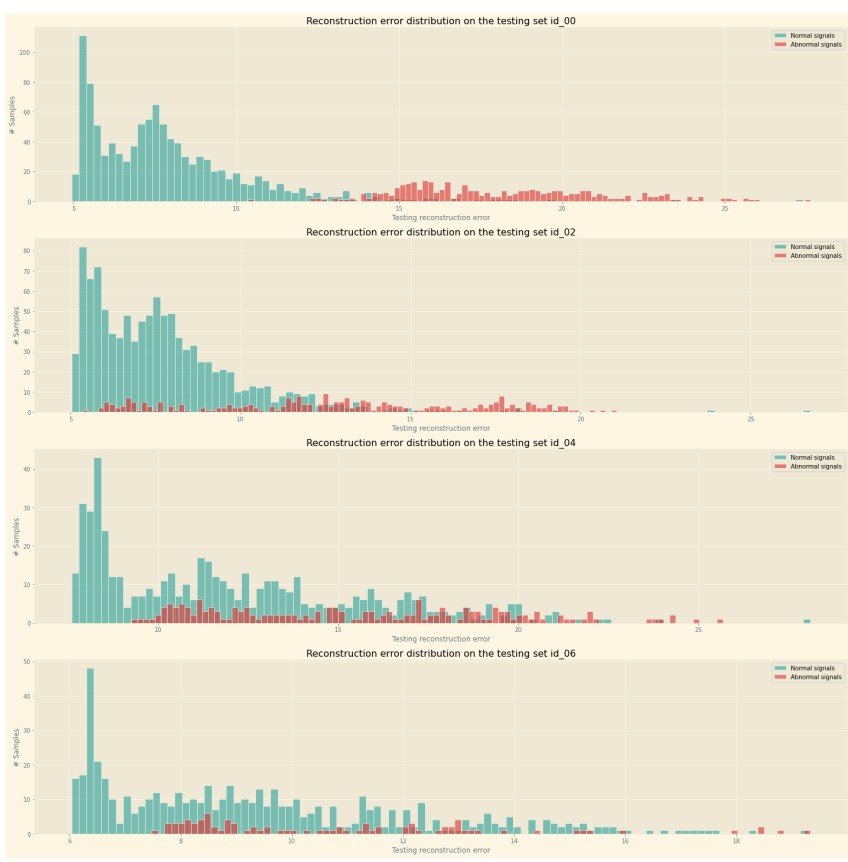

Figure 3: Histogram of reconstruction errors under normal and abnormal operation of machine id $00, 02, 04, 06$.

([AC17]). While changepoint detection has also been studied from a Bayesian perspective, much of this literature has focused on the batch setting (see [Gun+21; AM07; FL07]). Though recent works have begun to focus on Bayesian online detection, there has been little work (apart from [AMF20]) on proving finite time guarantees.

Our work is motivated by costs imposed on data collection due to resource constraints. In this context, while adopting a frequentist perspective, we propose methods for adaptive online data selection for multi-stream time-series data. Recent works have begun to focus attention on adaptive online data collection for changepoint detection from both frequentist [ZM20] and Bayesian [OGR10; Gun+21] perspectives. Additionally, while different from our focus, we mention in passing that methods for active change-point detection [HKK19], where the task is to adaptively determine the next input have also been proposed. Furthermore, there are a number of works that focus on bandit regret minimization for non-stationary time-series data [GM11; LLS18; MS13]. While these works are related we note that regret minimization is a fundamentally different objective from changepoint identification where the goal is to minimize detection delay.

We will outline similarities and differences between our work and closely related prior works. From a practical perspective, [OGR10] is similar to our work in that they too motivate their approach from a sensor network viewpoint, where sensors may undergo faults or changepoints exhibited due to environmental factors. They propose a Bayesian formalism largely based on the well-known Bayesian online change detection (BOCD) method [AM07], and leverage Gaussian processes (GP) for modeling. The GP perspective allows for tractable sequential time-series prediction and sensor selection. Recently, [Gun+21] have proposed to extend the BOCD approach to incorporate costs in making decisions for real-time data acquisition in multi-fidelity sensing scenarios. Different from these perspectives, our approach is frequentist, and does not impose action-specific costs and distributional constraints on the underlying latent parameters or on the changepoints. Furthermore, we derive finite time performance bounds, which is not a focus of any of these works.

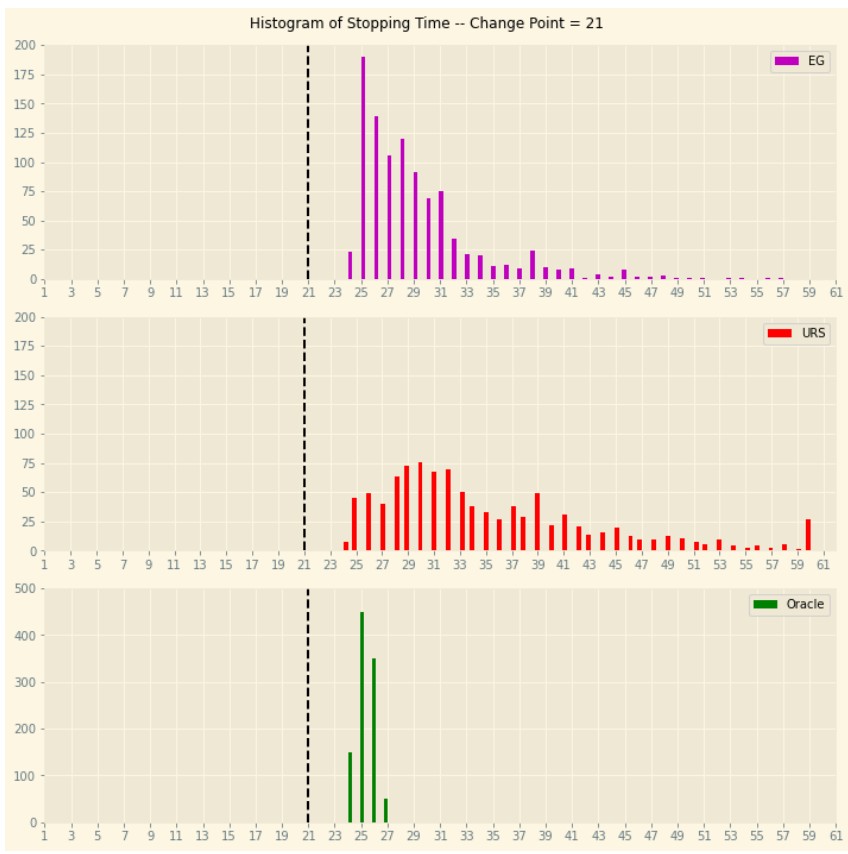

Figure 4: Audio based change detection of machine anomaly: Histogram of stopping times by URS, Oracle, and $\epsilon$-GCD (EG) for changepoint $\nu = 21$.

Similar to our work, [ZM20] also adopts a frequentist perspective and proposes a method for bandit changepoint detection based on the well-known Shiryaev-Roberts-Pollak scheme. To adaptively choose data streams/sensors, they utilize Thompson sampling [Tho33] to balance exploration of different data streams for acquiring knowledge with exploitation of informative streams. In addition, they present theoretical properties, and show that their method does not trigger false alarm too soon. However, detection delays under false alarm constraints are not explicitly characterized, which is a key challenge.

In summary, the principal difference between these prior works on bandit online changepoint detection and our $\epsilon$-GCD is that we are able to explicitly characterize information theoretic lower bounds on expected detection delay under a false alarm constraint. Furthermore $\epsilon$-GCD is natural variant of CUSUM, and our explicit analysis shows that it exhibits finite-time performance guarantees on expected detection delay and matches our lower bound at low false alarm rates. This leads to optimality that is absent in prior work.

## F    Discussion and Future Work

This work has laid down a principled approach to exploration with information-limited sensing to rapidly detect changes in distribution. Specifically, we have shown that relatively 'simple' (epsilon-greedy) forced exploration is sufficient to obtain detection delays comparable to an oracle who knows the post-change distribution beforehand.

As such, this study represents only an initial attempt to understand the limits of adaptive sensing for change detection, and opens up a host of interesting avenues for further investigation. These include (a) the possibility of 'more adaptive' exploration approaches, such as confidence-set or posterior sampling-based methods, that could improve the delay for learning a good guess of the post-change distribution (the second term in the detection delay bound), (b) adaptive sensing when

both the pre change and post change distribution is unknown, which also entails learning the default distribution online, (c) extensions to continuous parameter spaces, (d) detecting multiple changes that occur continually over time, and (e) studying the adaptive change detection problem for Markovian dynamics or controlled processes.