# OpenReview forum: "Bandit Quickest Changepoint Detection"
_NeurIPS.cc/2021/Conference — NeurIPS 2021 Poster_

### Official Review · Reviewer_r9YP · 2021-07-13

**Rating:** 6
**Confidence:** 3

**Summary:**

The paper considers the problem of minimising the delay in detection of a changepoint in the distribution of sequentially observed data, subject to a constraint on the rate of false alarms. In particular, this paper is focussed on a bandit-like version of the problem, where a decision-maker must choose what facet of the data to observe at each time point. An example of such a situation is where the sequentially observed data are multivariate, and the decision-maker may only observe one co-ordinate per time-point and must sequentially choose which co-ordinates to observe at which time points, with a view to accurately and rapidly detecting changes. The classical quickest changepoint detection problem may be recognised as a special case of this framework where the decision-maker always has the option to observe all dimensions of the data. The paper has three main contributions: i) a characterisation of the bandit quickest changepoint detection via a lower bound on the detection delay of any algorithm for a given upper bound on the desired false positive rate ii) an algorithm ($\epsilon$-GRD) for the bandit quickest changepoint detection problem, and iii) a theoretical guarantee on the performance of $\epsilon$-GRD, showing its desirably small detection delay and its order with respect to the magnitude of the change, structure of the action set, and desired false positive rate. The paper concludes with a numerical study of the algorithm on an open source dataset.

**Limitations And Societal Impact:**

considered sufficiently.

**Main Review:**

I am quite satisfied with the quality and significance of the theoretical and practical work, and would, overall recommend acceptance of the paper – but I would also recommend certain regards in which the presentation could be improved, in particular so as to make the nature of the problem clear from the outset.

ORIGINALITY: The authors do a good job of contextualising their work within the existing literature, drawing parallels to quickest changepoint detection and bandit best arm identification works. The contribution of the paper is original in the sense that the algorithm is new for the problem and that a finite-time analysis of the problem is novel. I am satisfied that the approach and analysis provide a sufficiently original contribution to the literature.

QUALITY: What I have read of the theoretical work is mostly sound, and I appreciate the provision of detailed proof sketches. The assertion that the argmax is a queue on line 197 is confusing to me. Could you provide some more detail here as to what is meant?

I read the Appendix containing theorem 1 in more detail and have two questions, the answers to which should perhaps be made clearer in the paper: on line 602, why is it the case that decisions are independent of the distribution? I would have expected this to have some affect for certain algorithms, certainly for $t>\eta$, in line 605, how has the $P^(\infty)(E)$ term been removed so that we arrive at the statement of Lemma 3. I’d also suggest in this section helping the reader out by more specifically referencing the result ported from [GMS19] – e.g. via an equation number of Theorem reference.

CLARITY: Generally the writing is accurate and clear. However I do feel there is work that could be done to present the setting more clearly from the outset. In particular, the description in the introduction is somewhat ambiguous. In the section beginning on line 28 it is not obvious what the nature of the actions in \mathcal{A} is likely to be – I appreciate the generality of the theory but it is perhaps intimidating in the first instance. It is similarly unclear if X is univariate or multivariate, or if it matters, how the decision-maker reports a change, whether there is a single change or multiple, and (later) why it is realistic to assume that the initial parameter is known. I think a concrete example used at this point (e.g. imagine a sensing network where data … is observed, and actions take the form of … etc) would be beneficial in several regards – providing an example to hang the thinking of, clarity in terms of what is meant by a sensing action etc, and motivating the relevance of the problem. I suspect a yet more useful situation would be to merge the problem specification in section 2 with the initial introduction and use the full precision from the outset in combination with comments as to the intuition, but this may be too much to promise in the rebuttal.

A couple of more minor points are that the equation in lines 193-4 would be much easier to read in an equation display environment, and that similarly Algorithm 1 would be better presented over the full page width.

SIGNIFICANCE: I am not expert in the technologies where this model and its solutions are likely to find use so am unqualified to speak confidently to the realism of the motivating examples. However, it seems very plausible that this bandit-like version of quickest changepoint detection should be a problem encountered in practice, and from a theorists point of view I found the problem and the paper interesting – so I would assess the paper as providing a suitably significant contribution.


**Time Spent Reviewing:**

5

---

> ### Author Response · Authors · 2021-08-08
> **Clarifications regarding Queue interpretation of specific equations**
>
> **Comment 1. The assertion that the argmax is a queue on line 197 is confusing to me. Could you provide some more detail here as to what is meant?**
>
> *Response 1.*
> A queue is just a (random) process $X_t$ that satisfies $X_{t+1} = (X_t + V_t)^+$, for some other random variables $V_1, V_2, \ldots$ which admit interpretation as 'arrivals' (they can be positive or negative). In other words, think of $V_t$ as the amount of work brought in by an arriving customer to the queue at time instant $t$. The queue is not allowed to hold a negative amount of work at any time by this definition. Since the process $Q_t^{(1)}$ on line 197 indeed admits such a recursion (as in line 198), it can be interpreted as a queue with arriving work equal to the log likelihood ratio $E_\ell \log \frac{f_{\theta}(X_\ell | A_\ell)}{ f_{\theta_0}(X_\ell | A_\ell) }$.
>
>
> **Comment 2. on line 602, why is it the case that decisions are independent of the distribution? I would have expected this to have some affect for certain algorithms, certainly for , in line 605, how has the  term been removed so that we arrive at the statement of Lemma 3.**
>
> *Response 2.*    A causal decision making algorithm is specified by what action it makes as a function of the history of  *observations* so far, and not their probability distribution. In other words, the description of an algorithm (think UCB in the multiarmed bandit regret problem, for e.g.,) lives independently of what distribution or environment the algorithm is operated in (the UCB algorithm's description does not change depending on what distribution is actually generating the rewards).
>
> Please see standard references such as https://tor-lattimore.com/downloads/book/book.pdf [Lemma 15.1] and https://www.jmlr.org/papers/volume17/kaufman16a/kaufman16a.pdf [Appendix A] in order to understand formally the notion of a decision-making algorithm and lower bounds based on change of measure.
>
> **Comment 3. I’d also suggest in this section helping the reader out by more specifically referencing the result ported from [GMS19] – e.g. via an equation number of Theorem reference.**
>
> *Response 3.* Certainly; this is precisely equation (11) from [GMS19].
>
> **Comment 4. (Clarity)description in the introduction is somewhat ambiguous. In the section beginning on line 28 it is not obvious what the nature of the actions in $\mathcal{A}$ is likely to be – I appreciate the generality of the theory but it is perhaps intimidating in the first instance; It is similarly unclear if X is univariate or multivariate, or if it matters, how the decision-maker reports a change, whether there is a single change or multiple, and (later) why it is realistic to assume that the initial parameter is known. I think a concrete example used at this point (e.g. imagine a sensing network where data … is observed, and actions take the form of … etc) would be beneficial in several regards – providing an example to hang the thinking of, clarity in terms of what is meant by a sensing action etc, and motivating the relevance of the problem.**
>
> *Response 4.*
> Thanks for expressing your confusion about this. See Lines see lines 372-384, 319-413, 250 and our response to Fs8y Comment 4 and Concrete sensor network example above. We will revise the paper and bring some of this discussion forward.
>
> **Comment 5. (practical motivation) am unqualified to speak confidently to the realism of the motivating examples.**
>
> *Response 5.* We designed Experiments (see lines 372-384 and 319-413), motivated by real-time anomaly detection in sensor networks (see OGR10 for weather sensor network, and examples such as Gun+21 Audio dataset). Our model of structured anomalies (see OGR10) is an instance of what we expect in a weather monitoring network exhibiting spatially-local temporal anomalies. See also Concrete Example.

---

> ### Author Response · Authors · 2021-08-10
> **Clarity: Concrete sensor network example that grounds notation, problem scenario, and sensing costs.**
>
> >Concrete examples with clearly enunciated observation and action spaces appear in Sec.~6 (see lines 372-384 and 319-413). We will describe this briefly here for reviewer reference, and bring this discussion forward in the revised version.
>
> **Sensor Network Example**
> >As a case in point, consider a sensor network composed of $N$ magnetometers in a spatial field, where the goal is to detect sudden appearance of targets. A reasonable model for the scenario is what is written on line 375. Observation at sensor $j$ follows: $S_j(t)=\theta_j \mathbf{1}_{t \geq \nu} + W_j(t)$; $S_j(t)$, $\theta_j$ and $W_n(t)$ are real-valued. $W_n(\cdot)$ is assumed to be IID zero mean Gaussian process.  A sensor observation exhibits elevated activity if there is change ($\theta_j > 0$), and this elevated activity manifests only among $K$ sensors within a small spatial range of the activity, i.e. the support of $[\theta_j]$ is a K-connected set (Connectivity is defined in terms of spatial proximity of the sensors). The support and the size of $\theta_j$'s and the time of change, $\nu \in \mathbb{R}$ are unknown. When the change happens at the unknown time $\nu$, at some location in the spatial field, the $K$ sensors in close proximity to that location, sense this change. Sensors outside this region do not observe changes. The post-change parameter set $\Theta^{(1)}$ is the set of all $[\theta_j]$ whose non-zero support forms a $K$ connected set. Our action set ${\cal A}$ depending on application can admit only pointy actions corresponding to single node observations, or a subset of streams (multi-node observations) or a mixture (linear supervision of node observations), and a query $a_t \in {\cal A}$ results in observation $X_t=\langle a_t, S(t)\rangle$.
>
> **Why it makes sense to assume initial parameter is known?**
> >This does assume that pre-change statistical models are known. However, this is reasonable since it can often be estimated because under normal working conditions we can collect sufficiently large data. Post-change distributions are more uncertain, for instance the location or the level of elevated activity is unknown ahead of time. Nevertheless, as seen above the post-change distributions can be readily parameterized.
>
> **What are sensing Costs?**
> >Our adaptive sampling setup accounts for sensor costs. It precisely models the requirement that we want only a fraction of ``energy-hungry'' sensors to be active for sensing and communication at any given time.
>
> **What is our goal?**
> >Our goal is to choose a sequence of actions from the action set, so that we minimize the detection delay (if change were to happen), and under the constraint that we do not falsely detect a change if no change ever happened. In this context, we tabulate results across different anomalous structures, diverse sensing actions, for both synthetic and real-world examples (see Supplementary and Main Paper).

---

> ### Comment · Reviewer_r9YP · 2021-09-02
> **Keeping my score**
>
> Hi authors,
>
> I will keep my score for this paper. Copied below is my justification for this from the earlier reviewers' discussion. Thank you for your time taken to reply to our comments.
>
> Having read the other reviews, and what I consider to be a convincing set of responses from the authors I'm inclined to keep my score as is. Since there are quite a lot of modifications to be made to the discussion/presentation with regards to our comments I would not want to go so far as marking as a clear accept, since I can't be sure of what the finished product will look like.

---

> > ### Author Response · Authors · 2021-09-03
> > **Modifications envisioned in the final version**
> >
> > We thank the reviewer for the feedback and kind consideration.
> >
> > While we agree that the paper will need to be revised, we would not go so far to say that these modifications are substantial. If one were to view this as a journal submission, the reviewer comments amount to what one would consider close to a minor revision. This is because the technical parts, experiments, related work, and supplementary remain the same. We reflected on our response and compared it to the current version of the paper. Here is a list of the type of responses we envision:
> >
> > 1. *Minor clarifications/typos.* (Fix broken reference on line 350, Make consistent use of $\theta_0$, Define $E_\ell$ formally before using it, $E_\ell$ and $A_\ell$ in line 197, Use threshold $\log \beta$ instead of $\beta$ in Alg. 1 for consistency with Thm. 2, Change $\tau$ to $(\tau -\nu)^+$ in Thm. 2, etc.)
> > 2. *Expanding text to highlight, clarify existing material.* This by far constitutes most of the modifications. For instance, reviewers asked for a concrete example to ground the problem in the beginning of the paper. This example already exists in entirety in the experimental section, and what this requires is to redo/move the example to introduction. In other parts, reviewers commented on specific aspects of the paper such as discussion about queue interpretation beyond citing a reference, and these types of exposition-level modifications require at most a  sentence or two.
> > 3. *Adding new additional material.* From what we understand, this is limited to adding the reviewer-proposed tradeoff curve.
> >
> > In summary, when we look at our responses, we were quite **liberal, generous and exuberant** in attempting to clarify reviewer questions. Indeed, in many places we could be done with citing specific lines in the text and proposing minor changes in revised version. Perhaps, the *open-review* system allowed us the luxury to be liberal. Nevertheless, we firmly believe the types of changes requested can be made well before the final submission deadline.
> >
> > Finally, note that we are allowed an extra page in the camera-ready version, and the additional content as listed above will easily fit within the allowance.
> >
> > In addition we will release the code for all the experiments, just as we have done for the peer review version here. This is part of our endeavor to provide a concrete code base for the research community for reproducible and thorough experimental comparisons to be made by future algorithmic advances in the area of bandit changepoint detection.

---

### Official Review · Reviewer_FMqm · 2021-07-16

**Rating:** 6
**Confidence:** 4

**Summary:**

This paper proposes a bandit quickest change detection framework under resource constraints for data collection. An adaptive online data selection framework is proposed by using a small amount of forced exploration along with greedy exploitation. Theoretical lower bounds to the detection delay are provided and the proposed \epsilon-GCD algorithm is proved to match the lower bound at low false alarm rates.

**Limitations And Societal Impact:**

Yes

**Main Review:**

The main contribution is the theoretical lower bound for the online change detection through adaptive data selection, and the near-optimality of the proposed epsilon-GCD algorithm. These are new additions to the bandit change detection theory and are consistent with classical theories.

It seems that there are some inconsistencies between the threshold \beta used in the algorithm and the theorems:
For theorem 2, the threshold \beta is not consistent with the \beta used in algorithm 1 -- in theorem 2, since the detection delay is roughly (log\beta)/KL, this means that the beta here refers to the threshold when using the likelihood ratio as detection statistic (without taking the log, as shown in the appendix), while the detection statistic in algorithm 1 is the cumulative log-likelihood ratio. In other words, the first term in detection delay should be beta/KL for threshold beta used in algorithm 1.

The numerical experiments are a bit weak: in the numerical results Table 1, better presents the mean and deviation for the simulated detection delay, and the same for Figure 1. Also, it would be interesting to see the tradeoff curves between the false alarm rate and the detection delay.

Minor issues:
(1) Line 197: should be sum E_\ell and X_\ell, A_\ell in the subscript; and E_\ell is not defined when introducing the likelihood in Line 193.
(2) Line 313: Line number missing (??) for Algorithm 1 mentioned
(3) Line 350: missing equation number (??)

**Time Spent Reviewing:**

3

---

> ### Author Response · Authors · 2021-08-09
> **Tradeoff Curves and correction so theorem Threshold and algorithm threshold are consistent.**
>
> **Comment 1. inconsistencies between the threshold $\beta$ used in the algorithm and the theorems: For theorem 2, the threshold $\beta$ is not consistent with the $\beta$ used in algorithm 1 -- in theorem 2, since the detection delay is roughly $(log \beta)/KL$**
>
> *Response 1.* This is a good catch. We inadvertently used the same variable name, and will fix it in the revised version. In words, if logarithms of likelihood ratios are used, as in Algorithm 1, then the threshold should be $\log(\beta)$, where $\beta = m \alpha^{-1} |\Theta^{(1)}|$. On the other hand if likelihood ratio is used then $\beta$ is the threshold. Just so we are on the same page, we will make the following modifications:
>
> 1. modify line 258 in Theorem 2 as follows:
> > (Time to false alarm) Let $\alpha \in (0,1)$ and $m \in \mathbb{N}$.
> >> If the stopping threshold $\log(\beta)$ is set as $\beta \geq \frac{m |\Theta^{(1)}|}{\alpha}$, then the stopping time satisfies  $\mathbb{P}^{(\infty)}(\tau < m) \leq \alpha$.
>
> 2. Similarly, modify line 4 in Algorithm 1 as follows:
> >**if** $\max Q_t^{(2)}(\theta) \geq \log(\beta)$ **then**
>
> 3. we will fix $Q_t^{(2)}$ in line 627 to be the logarithm of the likelihood instead of the likelihood.
>
> **Comment 2. The numerical experiments are a bit weak:tradeoff curves between the false alarm rate and the detection delay.**
>
> *Response 2.* Tradeoff curves appear here:
> > <https://www.dropbox.com/s/0yr8aijxkknzpyg/tradeoff_diff_scale.png?dl=0>
>
> There are two curves and our experimental setup corresponds to Table 1 in the main paper (pointy actions and isolated anomalies). Both curves are average values, averaged over 1000 Monte Carlo runs for varying values of $\beta$. Note that $\beta$ is $\Omega(\frac{m}{\alpha})$, where $\alpha$ is the false alarm rate, and thus serves as a surrogate for false alarms.
>
> > Red curve depicts stopping time under no change.  The tradeoff curve validates our theory in that the variation of stopping time is linear with respect to $\beta$. The stopping time here corresponds to the time when change is detected. As we increase the threshold $\beta$, the time when change is detected increases linearly with $\beta$. This is consistent with the first statement of Theorem 2.
>
> > Green curve depicts stopping time when change occurs at time $1$.  Here the green curve shows that the stopping time exhibits a logarithmic relationship, and this is consistent with the second statement of Theorem 2.

---

> > ### Comment · Reviewer_FMqm · 2021-08-27
> > **rebuttal acknowledgment**
> >
> > I thank the authors for the response and figures provided. Based on the comments and other reviews, I maintain my score.

---

### Official Review · Reviewer_P4K6 · 2021-07-23

**Rating:** 6
**Confidence:** 3

**Summary:**

This paper studies the bandit changepoint detection problem, where the observation ($X_t$) depends on multi-dimension action (sensing action ($A_t$)). The distributions associated with the problem change once at the unknown time $\nu$. The distributions before time $\nu$ are called pre-change distributions, and after $\nu$ are called post-change distributions. The goal is to design a policy to identify the change in the distributions as soon as possible.

The authors develop and analyze algorithms named $\epsilon$-Greedy Change Detector ($\epsilon$-GCD). Experimental results verify the performance of the proposed algorithm.

**Ethical Concerns:**

I do not find any ethical issue with the paper.

**Limitations And Societal Impact:**

Since this work is a theoretical paper, I do not find any direct negative societal impact.

**Main Review:**

Originality: A new algorithm $\epsilon$-GCD is proposed and analyzed for adaptive changepoint detection. The lower bound at a low false alarm rate for the changepoint detection problem is also given in the paper.
The authors have adequately cited the prior work and differentiated their work from previous contributions.

Quality: I find the results given in the paper technically sound, but I have not checked all the theoretical proofs in the Appendix.  Although the authors' claims are supported by theoretical analysis and experimental results, I find the following issues with the paper (depending on the authors' response, I may change my score):
1. Abstract (Lines 4-6) and Introduction (Lines 26-27) suggest that the authors want to minimize the sensing cost with detection delay. But they only focus on minimizing the detection delay (Line 34 and the way sensing actions are selected (Line 165) implies that the goal is to reduce the detection delay, whatever the sensing cost is).
2. Even though the distribution of the observations $(X_t)$ depends on sensing action $(A_t)$, but it is not clear how $X_t$ and $A_t$ are related. If there are no constraints, then they may be independent.
3. The pre-change distributions and possible post-change distributions (set) are known, which may not be possible in practice.

Clarity: The paper is well-organized, but the writing is needed to improve the readability of the paper. There are a few minor comments:
1. $E_l$ in Line 193 is used before defining it.
2. Line 197: subscript $t$ --> subscript $l$
3. Missing references in Line 313 (fixed in supplementary) and Line 350.


Significance: Other researchers may find some ideas in the paper useful for their future work on similar problems.

**Time Spent Reviewing:**

13

---

> ### Author Response · Authors · 2021-08-08
> **Clarification of Sensing Costs and Assumptions regarding pre and post change parameters**
>
> **Comment 1. Abstract (Lines 4-6) and Introduction (Lines 26-27) suggest that the authors want to minimize the sensing cost with detection delay. But they only focus on minimizing the detection delay (Line 34 and the way sensing actions are selected (Line 165).**
>
> *Response 1.*    By sensing cost we mean the number of sensors activated or queried at any given time. That is each sensor, when queried incurs a unit cost, and this cost is related to sensing the environment and communicating this information to the processing unit. The point of the paper is to develop theory for minimizing detection delays under false alarm constraints, under a fixed sensing budget, which amounts to the number of sensors that can be queried at each time. We will clarify this point in the revised paper. Note that to allow for multi-sensor queries, the action set ${\cal A}$ can be suitably enlarged to include a sensor subset, and in addition we could impose structural constraints on the subset (such as spatial proximity).
>
> We study the effect of varying sensing costs in the Experiments (see Supplementary Table 2 and 3 and our response to Review Fs8y Comment-4.), tabulating the effect of using diffuse or pointy sensing actions for structured changes in a spatio-temporal field. While our method imposes a fixed sensing budget in each round, one could consider overall average budgets. While this is outside the scope of our work, we believe that with the sequential independence condition on pre-change or post-change process in place, overall average budget constraint would not make much difference.
>
> **Comment 2. Even though the distribution of the observations depends on sensing action it is not clear how they are related. If there are no constraints, then they may be independent.**
>
> *Response 2.* First, our theory is agnostic to whether or not sensing action is dependent/independent of the observation. Indeed, if change is unobservable through any action in our action set, the KL divergences in the lower and upper bound are zero, and it should not be surprising that detection delay is unbounded. For instance, consider the linear observation model in line 250. Here we point out that $X_t \sim N(\langle a, \theta \rangle, \sigma_{a,\theta}^2)$. The independent case that reviewer has in mind arises when $\langle a, \theta \rangle=0$ for all $\theta \in \Theta^{(1)}\cup \{\theta^{0}\}$ and $\sigma_{a,\theta}^2$ is a constant. It is also clear how the observation is related to sensing in this model. For constant variance across sensors $\sigma\triangleq \sigma_{a,\theta}$, it is precisely the inner-product between the chosen action $a \in {\cal A}$ and the true parameter $\theta$ that describes the underlying observation process. In particular, analogous to linear bandits, we can write the regression equation: $E[X_t \mid \theta, a] = \langle a,\theta\rangle$. The difference is that in contrast to linear bandits, where $\theta$ is a fixed parameter, here $\theta$ can change to a new parameter at an unknown time. Furthermore, our goal in comparison to linear bandits is different. We want to locate the point of change rather than optimize cummulative rewards (i.e., $\sum_t \langle a_t, \theta\rangle$). We also refer to Fs8y Comment 2 for additional perspectives on difference between traditional bandits and BQCD. That said, we allow for general observation models with arbitrary relationships between actions and observations.
>
> **Comment 3. The pre-change distributions and possible post-change distributions (set) are known, which may not be possible in practice.**
>
> *Response 3.* Assumptions on pre-change are well motivated practically. See our response to Fs8y Comments 4 and 5, which contextualizes our contributions wrt prior works and provides motivation for our setup. See also Concrete Sensor Network example for R9YP.  We repeat salient points we make there.
>
> Consider again the sensor network for the scenario in Fs8y Comment 4 or Concrete sensor network example for R9yp. We point out that the problems we have at hand arise in practical problems such as local anomaly detection and intrusion detection.
>
> Such sensor network scenarios and models have appeared for the static case (see Sec 2 in Near-Optimal Detection of Geometric Objects by Fast Multiscale Methods, IT Transactions 2005 or Eq. 3 in Connected subgraph detection, AISTATS 2014). There is also a rich literature in conventional detection theory dealing with composite tests, where the null hypothesis is a known distribution, and the positive hypothesis belongs to a collection of parameterized distributions (see Introduction to Signal Detection, Poor, 1994, and Quickest change Detection, Poor 2009). Our framework can be viewed as bandit (or adaptive sampling) variant of that framework.
>
> As such, in our context, assuming knowledge of pre-change distribution is reasonable because we can collect sufficient data when the system is in normal operation. Post-change distribution is unknown because we may have different types of anomalies and we often do not have sufficient data for each anomaly type. Nevertheless, as described in Fs8y Comment 4, for the sensor network setting, it is still reasonable to assume that post-change (anomalous situation) process can be characterized by a parametric family. More generally, the fact that post-change process is characterized by a parametric family of distributions is quite general, and has strong roots in classical detection theory.

---

> > ### Comment · Reviewer_P4K6 · 2021-09-02
> > **Thank you for your responses**
> >
> > Dear authors,
> >
> > $~$
> >
> > I found your clarifications very clear. Accordingly, I have increased my score.
> >
> > $~$
> >
> > Regards,
> >
> > P4K6

---

### Official Review · Reviewer_Fs8Y · 2021-08-01

**Rating:** 5
**Confidence:** 3

**Summary:**

The paper provides a fundamental analysis of delay in the bandit change-point detection problem given a false alarm constraint. The authors propose a variant of CUSUM algorithm [Pag54], which is proven to asymptotically achieve the fundament limit of detection delay given well selected hyperparameter and a set of assumptions. Beside the theoretical analysis, they also provide interesting numerical experiments showing superiority over uniformly random selection strategy.

**Ethical Concerns:**

.

**Limitations And Societal Impact:**

.

**Main Review:**

It is clearly interesting to have such a fundamental limit analysis and algorithm achieving it. However, my major concern are the impractical assumption and the limited justification (theoretical and numerical), in particular, compared to existing ones as the considering problem is not new. I leave questions, comments and suggests in what follows.

- I believe the authors are also aware of exiting works showing optimality and optimal algorithm in similar bandit change-point detection problem. However, it is unfortunate that there is no clear comparison with them in neither theory nor experiment. At least, the algorithm in  [ZM20] seems comparable to the proposed algorithm.

- The framework of considering false alarm probability may be able to be converted/connected to the framework of piecewise-stationary bandit, ​e.g., Cao et al. "Nearly Optimal Adaptive Procedure with Change Detection for Piecewise-Stationary Bandit", 2019. Any discussion on these two frameworks would be helpful.

​- I don't think the example of cameras with local view is really motivating the proposed framework, in which $\theta^{(0)}$ and $\Theta^1$ are known in advance. Such a set of prior knowledge seems hard to be obtained in practice. In addition, there is no connection to the trade-off between the detection and sensor activity budget.

- There are a number of typos (e.g., $\theta_0$ or $\theta^{(0)}$) and editorial errors (especially, broken references). The notation system overuses super/sub-scriptions. I don’t understand what $(1)$ means in $\Theta^{(1)}$.

**Time Spent Reviewing:**

12

---

> ### Author Response · Authors · 2021-08-07
> **There is no connection between Cumulative Regret in Bandit Setting and Bandits in Quickest Change Detection even in non-stationary setting. So methods cannot be connected. This problem is new.**
>
> **Comment 1. Prior works show optimality and optimal algorithm in similar bandit change-point detection problem. no clear comparison with them in neither theory nor experiment.The problem is not new.**
>
> *Response 1*. We strongly disagree. Unfortunately, the comment conflates Bandits encountered in prior works such as multi-armed bandits (MAB) or linear bandits that seek to optimize cumulative regret with the bandit quickest change detection (BQCD) problem described here. We emphasize (see lines 86-89 and pg.1 footnote) MAB fundamentally bears no relationship to BQCD. We elaborate on this point in Comment 2.
>
> In fact, [ZM20] and [Gun+21], which are contemporaneous to our work, are the only ones dealing with our problem. Thus the problem is new. Furthermore, ours is the first paper to develop fundamental lower and upper bounds for BQCD, and characterize optimality in this context.
>
> **Comment 2. may be able to be converted/connected to the framework of Cao et al. "Nearly Optimal Adaptive Procedure with Change Detection for Piecewise-Stationary Bandit", 2019**
>
> *Response 2.* In view of Response 1, let us clarify the fundamental difference between BQCD and Cao et.al. 2019. Suppose we consider a MAB problem with a single arm, then, regardless of whether or not reward distribution is non-stationary, the cumulative regret is identically zero. On the other hand, for BQCD, although we now have full information, a strong false alarm constraint ($\alpha \rightarrow 0$)  that imposes negligible false alarms over the horizon $T$ leads to $O(T)$ detection delay, even when the pre and post change rewards distributions are well-separated (see lower bound in Theorem 1).
>
> More generally, there is no explicit notion of `reward' of an action that is part of the problem description here. Cao et.al. 2019 describe a method for minimizing regret in the face of non-stationary distributions, and, in fact, rely on classical (non-adaptive) change point detection procedures as a subroutine along the way to reward optimization. Our focus is on identifying when (if any) change happened assuming the ability to be adaptive. This point has also been elaborated in [ZM20]. In fact, considering general MAB in the non-stationary setting, we see that large errors in change-point detection, say $O(\sqrt{T})$ can still maintain $O(\sqrt{T})$ cumulative regret. However, this would not be satisfactory in BQCD.
>
> **Comment 3. no clear comparison with them in neither theory nor experiment. At least, the algorithm in [ZM20] seems comparable to the proposed algorithm.**
>
> *Response 3*. In view of points above our comparisons can only be wrt [ZM20,Gun+21]. [ZM20] is yet unpublished, and [Gun+21] adopts a Bayesian viewpoint (namely, changepoint and post-change parameters are drawn by Nature from a prior and metrics are averaged w.r.t it). That said, [ZM20] does have some analysis but their results are too weak to merit a comparison to our work. [ZM20] shows that ``in principle'' their method will control false alarms for sufficiently large thresholds, and independent of false alarms would eventually choose sensors where changes manifest. In contrast, we derive fundamental detection delay under a false alarm rate constraint, thus fully characterizing the BQCD problem.
>
> *Experiments*.
> This being a theory paper, our goal for the experiments was to expose the theoretical results. In this context, we experimented with the audio dataset from [Gun+21], and investigated the impact of structured and unstructured spatio-temporal anomalies in the context of fine-grained and diffuse action sets. These experiments depict the role of diverse observation models on performance.
>
> *Applicability of [ZM20]*.
>
> There are also differences in terms of applicability of the method for our setting. (a) [ZM20] assumes independent sensors and knowledge of pre and post-change parameters. This is an issue because this does not handle correlated or structured anomalies, which is a significant aspect of our part of our experiments; (b) [ZM20] has a number of tuning parameters, the code is not published, and we were not confident in our implementation; (c) [ZM20] in its generic embodiment (distribution $G$ is a point mass--see sec 3.1.2) without these tuning parameters is in essence a pure exploitation approach, and it is unclear that this would work. This makes direct comparison difficult.
>
> **Comment 4. impractical assumption and the limited justification (theoretical and numerical)**
>
> *Response 4*. We strongly disagree. In point-of-fact, we designed Experiments (see lines 372-384 and 319-413), motivated by real-time anomaly detection in sensor networks (see OGR10 for weather sensor network, and examples such as Gun+21 Audio dataset). Our model of structured anomalies (see OGR10) is an instance of what we expect in a weather monitoring network exhibiting spatially-local temporal anomalies.
>
> *Example Scenario*.
> As a case in point, consider a sensor network composed of magnetometers in a spatial field (see response for r9YP for detailed description). A reasonable model for the scenario is what is written on line 375. Observation at sensor $j$ follows: $S_j(t)=\theta_j \mathbf{1}_{t \geq \nu} + W_j(t)$. A sensor observation exhibits elevated activity if there is change, and this elevated activity manifests only among the sensors within a small spatial distance of the activity. This translates to the assumption that the support of $[\theta_j]$ is a connected set of $K$ proximal nodes, which are a priori unknown to the learner. The support, the size of $\theta_j$'s and the time of change are unknown. In the static (no temporal aspect) setting this is a standard model (see see Sec 2 in Near-Optimal Detection of Geometric Objects by Fast Multiscale Methods, IT Transactions 2005 or Eq. 3 in Connected subgraph detection, AISTATS 2014).
>
> This does assume that pre-change statistical models are known. However, this is reasonable since it can often be estimated because under normal working conditions we can collect sufficiently large data. Post-change distributions are more uncertain, for instance the location or the level of elevated activity is unknown ahead of time. Nevertheless, as argued below (also see Concrete Sensor Network example for r9yp and comment 3 for P4K6), the post-change distribution can be readily parameterized.
>
> Our adaptive sampling setup accounts for sensor costs. It precisely models the requirement that we want only a fraction of ``energy-hungry'' sensors to be active for sensing and communication at any given time. In this context, we tabulate results across different anomalous structures, diverse sensing actions, for both synthetic and real-world examples (see Supplementary and Main Paper).
>
> **Comment 5. Theoretical Assumption is strong, and limited wrt prior works. $\theta^{(0)}$ and $\Theta^{(1)}$ cannot be known in practice.**
>
> *Response 5*. We disagree. In view of our example scenario in Response 4, it is reasonable to assume knowledge of pre-change distribution ($\theta^{(0)}$). Assuming that the anomalous process after the change is unknown but takes values in a parameterized set is not only reasonable, but it arises naturally. In the example scenario, this would be the collection of all K-proximal nodes among the $N$ nodes in the sensor network. In addition the mean values on each element in the collection can be additionally parameterized to incorporate uncertainty in the amount of change.
>
> Furthermore, our theoretical assumption is more general than any of the prior works [ZM20,Gun+21] and accounts for scenario described above (see Comment 3). Second, the fact that we assume that the statistical model before the change is well-characterized is not only standard in existing literature but also reasonable because one typically has sufficient data to calibrate the system under normal operating conditions. Indeed, this is standard in conventional batch anomaly detection problems (see One-class SVMs) but also assumed in [ZM20,Gun+21].

---

### Decision · Program_Chairs · 2021-09-28

**Decision:**

Accept (Poster)

**Comment:**

This is a borderline paper. After thorough discussions among Reviewers, I would lean towards a rejection.

Reviewers have posted many excellent advices which I hope the authors would be able to take advantage of in their revision. The following additional comments are my own read:

The paper considers distributed quickest section, where the trade-off between sensing cost and detection delay is optimized in a bandit feedback setting. First, an information theoretic lower bound is derived on the detection delay for parameterized probability distributions. Then a computationally simple sensing scheme  that has optimal expected detection delay at low false alarm rates is proposed, and the empirical performance is tested on real and synthetic datasets.

Some of the results, especially the lower bounds for distributed detection are interesting.  Some concerns are raised below:

#. The framework is not clearly defined. There should be a trade-off between the cost of information acquisition and the informativeness of the signal received. This is a natural trade-off. Perhaps a trade-off between number of sensor pulls (crude measure of informativeness) and detection delay would have made for a clearer framework.

#.	The reason for not using regret as a performance measure is poorly motivated. This makes the problem very similar to classical change detection with a known likelihood, and renders the contribution incremental. This perhaps weakens the contribution of the lower bounds as well, as [Lai98] already establishes a lower bound in a similar setting.

#.	It is not clear what the dimension of $\theta$ is? It seems just pulling one arm alone provides information as to whether there is a change or not? Why bother exploring? Also, It is not clear what `more informative’ means here. Are there better sensors in the group that always provide more information?

#.	In step 13, the parameter is calculated using an MLE based on the whole history. This makes the policy to choose the `action’ in step 14, non-indexable. How is this addressed?

#.	Having a common parameter for all arms means that pulling any arm would provide information about all arms. How is the correlation information used?

Some other concerns:

#. Parametric quickest detection procedures, while interesting theoretically, are a serious limitation in an applied bandit framework, where the distributions are usually unknown or chosen by an adversary. How to extend the results to a non-parametric/ or perhaps a semi-parametric setting? To an adversarial setting?

#. Can $epsilon-$QCD be used for detecting transients?

$. Why is there no comparison with simple non-parameteric detection algorithms?

**Consistency Experiment:**

NeurIPS has a long history of experimentation. In 2014, NeurIPS ran an experiment in which 10% of submissions were reviewed by two independent committees to quantify the randomness in the review process. This year, we repeated a variant of this experiment to see how the quality of the review process has changed over time.  This paper was part of the experiment and was therefore assigned to two committees (consisting of reviewers, an Area Chair, and a Senior Area Chair) that reached independent decisions.  If both committees made the same recommendation, this recommendation was followed. If a single committee recommended acceptance, the paper was accepted (with the exception of a few cases in which the other committee identified what we considered a fatal flaw, e.g., an error in a key result).

This copy’s committee reached the following decision: **Reject**

The other committee assigned to the paper recommended **Accept (Poster)**.  You can find the other set of reviews, along with any follow up discussion with the authors here:
https://openreview.net/forum?id=mxowVJFe8D5